# Machine learning for improved data analysis of biological aerosol using the WIBS

**Simon Ruske[1], David O. Topping[1], Virginia E. Foot[2], Andrew P. Morse[3], and Martin W. Gallagher[1]**

[1]Centre of Atmospheric Science, SEES, University of Manchester, Manchester, UK
[2]Defence, Science and Technology Laboratory, Porton Down, Salisbury, Wiltshire, SP4 0JQ, UK
[3]Department of Geography and Planning, University of Liverpool, Liverpool, UK

**Correspondence:** simon.ruske@postgrad.manchester.ac.uk

**Abstract.** Primary biological aerosol including bacteria, fungal spores and pollen have important implications for public health and the environment. Such particles may have different concentrations of chemical fluorophores and will respond differently in the presence of ultraviolet light, potentially allowing for different types of biological aerosol to be discriminated. Development of ultraviolet light induced fluorescence (UV-LIF) instruments such as the Wideband Integrated Bioaerosol Sensor (WIBS) has allowed for size, morphology and fluorescence measurements to be collected in real-time. However, it is unclear without studying instrument responses in the laboratory, the extent to which different types of particles can be discriminated. Collection of laboratory data is vital to validate any approach used to analyse data and ensure that the data available is utilised as effectively as possible.

In this manuscript a variety of methodologies are tested on a range of particles collected in the laboratory. Hierarchical Agglomerative Clustering (HAC) has been previously applied to UV-LIF data in a number of studies and is tested alongside other algorithms that could be used to solve the classification problem: Density Based Spectral Clustering and Noise (DBSCAN), k-means and gradient boosting.

Whilst HAC was able to effectively discriminate between reference narrow-size distribution PSL particles, yielding a classification error of only $1.8\%$, similar results were not obtained when testing on laboratory generated aerosol where the classification error was found to be between $11.5\%$ and $24.2\%$. Furthermore, there is a large uncertainty in this approach in terms of the data preparation and the cluster index used, and we were unable to attain consistent results results across the different sets of laboratory generated aerosol tested.

The lowest classification errors were obtained using gradient boosting, where the misclassifiation rate was between $4.38\%$ and $5.42\%$. The largest contribution to the error, in the case of the higher misclassification rate, was the pollen samples where $28.5\%$ of the samples were incorrectly classified as fungal spores. The technique was robust to changes in data preparation provided a fluorescent threshold was applied to the data.

In the event that laboratory training data in unavailable, DBSCAN was found to be a potential alternative to HAC. In the case of one of the data sets where $22.9\%$ of the data was left unclassified we were able to produce three distinct clusters obtaining a classification error of only $1.42\%$ on the classified data. These results could not be replicated for the other data set where $26.8\%$ of the data was not classified and a classification error of $13.8\%$ was obtained. This method, like HAC, also appeared to be heavily dependent on data preparation, requiring a different selection of parameters depending on the preparation used. Further analysis will also be required to confirm our selection of the parameters when using this method on ambient data.

There is a clear need for the collection of additional laboratory generated aerosol to improve interpretation of current databases and to aid in the analysis of data collected from an ambient environment. New instruments with a greater resolution are likely to improve on current discrimination between pollen, bacteria and fungal spores and even between different species, however the need for extensive laboratory data sets will grow as a result.

# 1   Introduction

Biological aerosol, such as bacteria, fungal spores and pollen have important implications for public health and the environment (Després et al., 2012). They have been linked to the formation of cloud condensation nuclei and ice nuclei which in turn may have important influence on the weather (Crawford et al., 2012; Cziczo et al., 2013; Gurian-Sherman and Lindow, 1993; Hader et al., 2014; Hoose and Möhler, 2012; Möhler et al., 2007). These particles have impacts on health (Kennedy and Smith, 2012), particularly for those who suffer from asthma and allergic rhinitis (D'Amato et al., 2001). It is therefore of paramount importance that we continue to develop methods of detecting these particles, to quantify them, determine seasonal trends and to compare different environments.

There are a wide range of biological molecules, commonly referred to as biological fluorophores, that are known to re-emit radiation upon excitation e.g. amino acids, coenzymes and pigments (Pöhlker et al., 2012, 2013). Ultraviolet-light induced fluorescence (UV-LIF) spectrometers, such as the wideband integrated bioaerosol spectrometer (WIBS) have received increased attention in recent years as a potential methodology for detecting biological aerosol (Kaye et al., 2005). The WIBS uses irradiation at 280nm and 370nm to target some of the most significantly fluorescent bioflorophores such as tryptophan (an amino acid) and NADH (a coenzyme). These measurements are combined with an optical measurement of size and shape to further aid in discrimination.

Measurements from the WIBS have limited application in isolation. However, there are a range of techniques that could be used to predict quantities of biological aerosol from these fluorescence, size and morphology measurements. Techniques that could be used to solve this classification problem, include field specific techniques such as ABC analysis (Hernandez et al., 2016) as well as supervised and unsupervised machine learning techniques that are broadly used (Friedman et al., 2001).

It is not clear at this point what approach is preferred as all approaches have a range of advantages and disadvantages.

Supervised machine learning uses data collected within the laboratory, where the correct classification is known. Data is split into training data and testing data where the training data is used to fit a model which is then validated on the test set. Once a model is fitted and validated it may then be applied to classify ambient data.

During unsupervised analysis, ambient data is classified without using laboratory training data. Instead, an attempt is made to naturally segregate the data. Ideally, we may expect data to naturally be segregated into broad biological classes or into different groups of similar bacteria, fungal material and pollen, but this may not necessarily be the case.

The supervised methods, have the disadvantage that training data collected may not include the entirety of what might be collected during an ambient campaign. Particularly, in an urban environment, the instrument may collect measurements for a large quantity of non-biological material that should be classified as such or removed from the analysis. We would expect most of this non-biological material to either be non-fluorescent or weakly fluorescent and therefore it should be removed prior to analysis by applying a justifiable threshold to the fluorescent measurements (see Section 2.2). Nonetheless, a few weakly fluorescent non-biological particles may remain and could be overlooked if the training data is incomplete.

There are likely to be issues to be explored with either approach and therefore it seems unlikely that either supervised or unsupervised techniques can justifiably be abandoned at this point in time and it may well be the case that usage of a variety of techniques may be required to better understand the atmospheric environment. Nonetheless, it is still vital to investigate how these different techniques behave when analysing laboratory data to better understand how they can be most appropriately applied to ambient data.

In an ambient setting, determining the number of clusters is difficult, so Hierarchical Agglomerative Clustering (HAC) has been the preferred method over other methods such as k-means since the method naturally presents a clustering for all possible number of clusters (Robinson et al., 2013). A suggestion of the number of clusters can then be provided using indices such as the Caliński Harabasz Index (CH Index) (Caliński and Harabasz, 1974) by maximising a statistic which yields a peak for clusterings which contain clusters that are compact and far apart. HAC has previously been used on data collected using the WIBS to discriminate between different Polystyrene Latex Spheres (PSLs) and has been applied to ambient measurements collected as part of the BEACHON RoMBAS experiment (Crawford et al., 2015; Gallagher et al., 2012; Robinson et al., 2013).

Nonetheless relatively few studies have studied the usage of HAC on data from the WIBS (Savage et al., 2017; Savage and Huffman, 2018) . Evaluating the effectiveness of HAC on generated aerosol is crucial to support or repudiate conclusions made using HAC on ambient data, especially since the fluorescence response from the laboratory generated aerosol will much better reflect fluorescence responses from the environment, when compared with PSLs.

During the process of HAC there are also a number of vital choices that have to be made, that could have a substantial implication on the effectiveness of the method (these are discussed in detail in Section 2.2). For the PSLs previously analysed (Crawford et al., 2015), we determined standardising using the z-score, with removal of non-fluorescent particles, taking logarithms of shape and size was most effective. The CH index was selected to determine the number of clusters as it was demonstrated to perform best in the literature (Milligan and Cooper, 1985). It is however, not clear whether these choices will remain the most effective for lab-

oratory generated aerosol nor ambient data. See Section 2.3 for further details on data preparation for HAC.

Furthermore, data analysis using HAC can take a matter of hours, if not days depending on the number of particles. The time requirements for HAC are between $N^2$ and $N^3$ meaning that a doubling of the number of particles will require between four and eight times as much time. Such time requirements mean that not only is the method already quite slow, but will get increasingly slower as more data is collected, which may limit the real time effectiveness of the method.

Within the Python programming language, a package called Scikit-learn (Pedregosa et al., 2011) offers implementations of several unsupervised methods. Some of these methods i.e. Affinity Propagation, Mean-shift, Spectral Clustering and Gaussian mixtures are not explored as they will scale poorly as the number of particles increases (Pedregosa et al., 2011). Instead, our analysis is focused on K-means, HAC and DBSCAN which can be used on larger data-sets.

For HAC we continue to use the fastcluster package (described in Section 2.3). Sci-kit learn does have a HAC implementation but it is not as fast or memory efficient. We do use sklearn for DBSCAN and kmeans, although if one was to use DBSCAN for ambient data we would suggest exploring alternatives such as ELKI (Schubert et al., 2015) as the sci-kit learn implementation of DBSCAN by default is not memory efficient making it difficult to utilise for more than 30,000 particles. Sci-kit learn has a fast implementation for Gradient Boosting, so this is used.

## 2 Methods

In this section we discuss the variety of approaches that could be used to classify particles such as bacteria, fungal spores or pollen. In Section 2.1 we provide an overview of the instrument used to collect the data. In Section 2.2 we discuss the variety of decisions that need to be made prior to passing the data to the machine learning algorithms which are discussed in Sections 2.3 - 2.6. An overview of the different methods is given in Figure 1.

### 2.1 Instrumentation

The Wideband Integrated Bioaerosol Sensor (WIBS) collects size, shape and fluorescence measurements (Kaye et al., 2005). The size is a single measurement; the shape measurement consists of four measurements (one for each quadrant) which are combined to produce a single asymmetry factor measurement. A more precise definition of asymmetry factor has been provided previously in the literature (Gabey et al., 2010).

To measure fluorescence, the particle is irradiated with UV light at 280nm and 370nm from the firing of two xenon sources. Fluorescence emission is collected via two collection channels in the ranges $310 - 400nm$ and $420 - 600nm$.

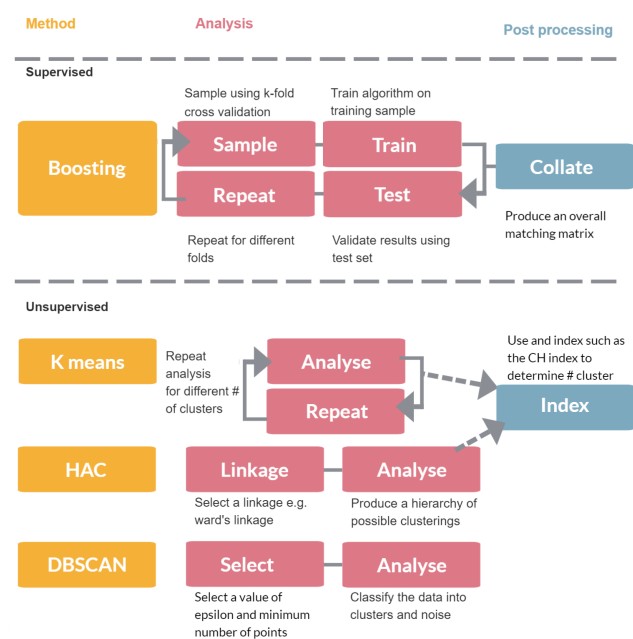

**Figure 1.** Overview of different analysis approaches

The 370nm xenon radiation lies within the first detection range and hence elastically scattered light from the particle, sufficient to saturate the detection amplifier, is received. This signal is therefore discarded.

After removal of this fluorescent measurement, there are three remaining fluorescence measurements. The notation FL1_280 is used to denote the measurement in the first detection channel when the particle is irradiated with ultraviolet light at 280nm and FL2_280 and FL2_370 are used to denote the measurements in the second detection channel when the particle is irradiated with ultraviolet light at 280 and 370nm respectively. These fluorescence measurements are combined with the size and asymmetry factor measurements. A more detailed description of the instrument can be found in previous publications (e.g. Gabey et al., 2010; Healy et al., 2012a)

### 2.2 Data preparation

Prior to analysis using the machine learning algorithm we may choose to make a variety of decisions to pre-process the data with the aim to improve performance (see Figure 2). An overview for the decisions often made are outlined below.

First we may elect to remove particles which are non-fluorescent. Forced trigger data is collected which is a measurement of the instrument response when particles are not present. We then set a threshold, for which if a particle fails to exceed this threshold in at least one of the fluorescent channels we conclude that the particle is non-fluorescent. Usually we set the threshold to be three standard deviations above the average forced trigger measurement although a recent labo-

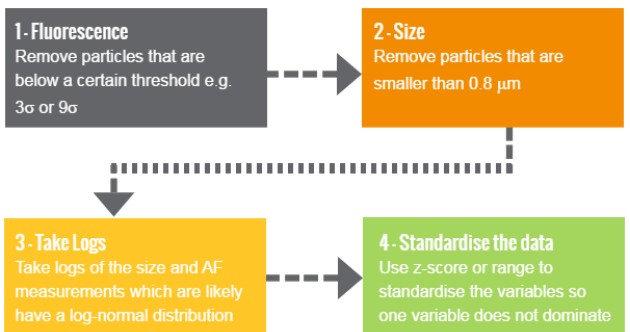

**Figure 2.** Overview of preprocessing steps for WIBS data

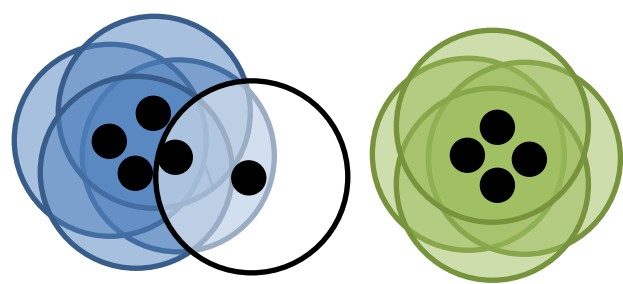

**Figure 3.** Visual representation of DBSCAN. Here each point is represented as a black dot and it's neighbourhood is represented by a circle. Here $\epsilon$ is the radius of the circle and the minimum number of points is 3. Four points have each been placed into the blue cluster and green cluster, all of which having at least 3 other points in their neighbourhood. One point is classified as noise as it has only 1 other point in it's neighbourhood.

ratory study has suggested that nine standard deviations may be more appropriate (Savage et al., 2017).

Another threshold is usually then applied to the size. A size threshold of $0.8\mu$m is usually applied as detection efficiency of the instrument drops below $50\%$ at this point. (Gabey, 2011; Gabey et al., 2011; Healy et al., 2012b).

Natural logarithms of the size and the asymmetry factor are often taken as these measurements are often log normally distributed and it is postulated that this will increase performance in the case of hierarchical agglomerative clustering.

It is also widely regarded that standardising the data prior to analysis is utmost importance (Milligan and Cooper, 1988). We often subtract the average measurement in each of the five variables and divide by the standard deviation, often referred to as 'standardising using the z-score'. Standardisation is used to prevent variables with larger magnitude, such as the fluorescent measurements, from dominating the analysis. An alternative approach to standardising is to divide each of the five variables by the range.

## 2.3 Hierarchical Agglomerative Clustering

In order for particles to be clustered, we need to define a measurement of how similar two clusters are. These similarity measures are often referred to as linkages. We use the Python package fastcluster (Müllner, 2013) which provides modern implementations of single, complete, average, weighted, Ward, centroid and median linkages (Müllner, 2011). A thorough detailing of the definitions of the different linkages can be found in the fastcluster manual (Müllner, 2013). For the memory efficient mode, which is essential when using the algorithm for large data sets, only Ward, centroid, median and single linkages are available.

Initially each particle is placed into an individual cluster. Next, using the linkage selected, the two most similar clusters are merged. The merging process is repeated until all the particles are placed in a single cluster, which provides a clustering from $k = 1, \cdots, N$, where $k$ is the number of clusters and $N$ is the number of particles being analysed. A cluster validation index such as the Calinski-Harabasz index (Caliński and Harabasz, 1974) is then used to identify an appropriate number of clusters. The index is maximised for clusterings that contain compact clusters that are far apart.

## 2.4 K-Means Clustering

K-Means clustering is designed to place particles into $k$ clusters. However we can repeat the method multiple times e.g. for $k = 1, 2, \cdots, 10$, where $k$ is the number of clusters. Similar to HAC we can then use a cluster validation index to determine which choice of $k$ gives the most effective results.

The method works as follows. Initially $k$ cluster centroids are set by selecting $k$ particles at random. The rest of the particles are then placed into these $k$ clusters depending on which of the centroids the particle is closest to. At this point a new centroid is calculated for each cluster. The process is then repeated many times until convergence occurs and the centroids do not change significantly from one iteration to the next.

## 2.5 DBSCAN

For DBSCAN we set two parameters, the radius for a neighbourhood $\epsilon$, and the number of particles required for a neighbourhood to be identified as dense.

Initially a random point, say A, is selected. If there are sufficient number of points in the neighbourhood of A then all the points in A's neighbourhood are also checked and so on, until the cluster has fully expanded and there are no points left to check. Should the point not have a sufficient number of other points in its neighbourhood then it is left unclassified. Further points are then selected and the above process is repeated until all points have been considered.

We give an example of DBSCAN in Figure 3. Note that cluster validation indices are *not* required for DBSCAN,

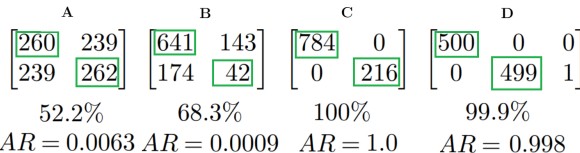

**Figure 4.** Four example matching matrices. Immediately below each matrix is the percentage of particles placed into the same cluster for both clustingers in each case. At the very bottom we have the adjusted rand score.

since the number of clusters is intrinsically calculated within the algorithm.

## 2.6 Gradient Boosting

A basic decision tree is constructed by considering each possible split across all variables and evaluating which split best divides the data. For example, we may consider the third fluorescence channel and split the data on the basis of whether the measurement is more or less than 10 arbitrary units (AU). This process is then repeated many times until a tree is built.

There are two ways in which trees can be combined into an ensemble. The first is by averaging multiple trees in the hope to produce a more accurate classification as is the case in random forests and bagging classifiers (Breiman, 2001, 1996). In the case of random forests and bagging the data set is sampled with replacement, meaning that the same particle could be selected more than once or not at all. Sampling in this way enables the algorithm to produce a subtly different version of the data from which to build each tree. In addition, when using a random forest, instead of considering all possible variables to use to split the data, only a random subset is used.

Alternatively we can fit a single decision tree to the data, evaluate where the tree is performing well and then fit a second tree to the particles in the data for which the current model is performing poorly. This process can be repeated many times, each time adding a new tree to the model in the hope of making an improvement. This approach is known as AdaBoost (Freund and Schapire, 1997). Gradient Boosting is an extension of AdaBoost to allow for other loss functions (Friedman, 2001).

For the current study we elect to use Gradient Boosting to indicate the performance of the supervised approach since it was the best performer for the Multiparameter Bioaerosol Spectrometer, a similar UV-LIF spectrometer similar to the WIBS but with single waveband fluorescence, 8 fluorescence detection channels and very high shape analysis capability (Ruske et al., 2017)

## 2.7 Evaluation Criteria

To aid in evaluating how well methodologies performed we used two tools: the matching matrix (Ting, 2010) and the adjusted rand score (Hubert and Arabie, 1985).

In Figure 4 we present four different matching matrices. To produce these matrices we compared: two random clusterings with approximately 50% of the data in each cluster (A); two random clusterings each with 80% and 20% of the data in each of the two clusters respectively (B); two identical clusterings (C); and two clusterings which were nearly identical except one data point had been placed into a third cluster for one of the clusterings.

### 2.7.1 Matching Matrix

The matching matrix, often referred to as a confusion matrix, can be used as an aid in comparing two clusterings.

In the case of the current manuscript, we use this to compare the output from an algorithm with labels assigned to each particle. We may assign labels to indicate what broad type the particle is (e.g. 1 if the particle is bacteria, 2 if the particle is fungal etc.) or we may assign labels to indicate what sample a particle is from (e.g. 1 if the particle is *Bacillus atrophaeus*, 2 if the particle is *E. coli* etc.)

Consider example C in Figure 4. This matching matrix compares two clusterings each containing two clusters. Each row corresponds to a cluster in the first clustering and each column corresponds to a cluster in the second clustering. The element in the first row and the first column (in this case 784) indicates the number of particles that were placed into the first cluster in the first clustering that were also placed into cluster 1 in the second clustering. Two identical clusterings will produce a matching matrix that has non-zero values only the diagonal.

A and B in Figure 4 are examples of poor performance and C and D are examples of very good performance.

### 2.7.2 Adjusted Rand Score

When evaluating a large number of clusterings, it may be useful to use a statistic to summarise the information in the matching matrix. In a previous study (Ruske et al., 2017), we used percentage of particles correctly classified as a statistic for indicating performance. This is an easy to interpret statistic, but can be misleading when used on imbalanced data. In both example A and B, we have two randomly generated clusterings. However in B we have 80% of the data points placed into the first cluster, whereas in A the data points are approximately equally distributed between the two clusters. The percentage of points which are placed into the same cluster for both clusterings are 52.2% and 68.3% for A and B respectively. We can see that the more imbalanced a data set is, the more likely data points are to be placed into the same clusters. It is for this reason we elect to use an alternative

statistic: the adjusted rand score. This statistic attains a value of approximately zero for both A and B.

Comparing clusterings is a developing area of research and there are other alternative statistics such as the mutual infor-
5 mation score (Vinh et al., 2010) that could be preferable to the adjusted rand score. However our initial tests (not presented), indicated that calculation of the mutual information often required an order of magnitude more time than the calculation of the adjusted rand. Therefore we elected to use the
10 adjusted rand score for the current study.

## 3   Data

The efficacy of the different data analysis approaches was evaluated using three different data sets. The first of which comprised several industry standard polystyrene latex
spheres of various different sizes and colours. This data set was first analysed in Crawford et al. (2015), where Hierarchical Agglomerative Clustering was successfully applied to the data yielding a classification accuracy of $98.2\%$. This data set presents a simple challenge for which we would expect any
reasonable algorithm to be able to discriminate between the different sizes and colours of particles.

To further extend the previous analysis in Crawford et al. (2015) we include two data sets collected in 2008 and 2014 which are similar to data previously published using the Mul-
25 tiparameter Bioaerosol Spectrometer (Ruske et al., 2017). A subsection of the data collected 2014 has previously been analysed in the appendix of (Crawford et al., 2017). These data sets consist of various different pollen, fungal, bacterial and non-biological samples, and should present a much more
difficult challenge for the algorithms.

The samples of laboratory generated aerosol were collected as follows. Material was aerosolised into a large, clean HEPA filtered chamber, which incorporated a recirculation fan. The *Bacillus atrophaeus* and *Escherichia coli* (*E.coli*)
bacteria were aerosolised into the chamber using a mininebuliser (e.g. Hudson RCI Micro-Mist nebuliser) as were the salt and phosphate buffered saline samples. The dry samples, which included the pollen, and fungal samples were aerosolised directly into the chamber from small quantities
of powder utilising a filtered compressed air jet. The diesel smoke and grass smoke samples were generated by burning a small amount within a fume cupboard using a smoker (piece of bespoke equipment). The bacterial samples were either washed or unwashed and diluted or undiluted.
We present a summary of the number of particles for each sample after a fluorescent threshold of $3\sigma$ and $9\sigma$ is applied in Tables 1, 2 and 3. In 2008 the thresholds are constructed using forced trigger data collected at the same time as the experiment, whereas in 2014 the thresholds are constructed
using forced trigger data collected using the same instrument at an earlier date. Ideally, the threshold for the data collected in 2014 would be constructed using forced trigger data col-

**Table 1.** The number of particles remaining after a fluorescent threshold of $3\sigma$ or $9\sigma$ was applied for each of the bacterial samples collected in 2008. Each sample was either washed or unwashed and diluted or undiluted. Washed samples are denoted by a check mark in the column "W" and diluted samples are mark in the column 'Dil.'.

| ID | Sample | W | Dil. | $n > 3\sigma$ | $n > 9\sigma$ |
|---|---|---|---|---|---|
| A | *Bacillus atrophaeus* Spores | | | 952 | 34 |
| B | | | ✓ | 52 | 4 |
| C | | ✓ | | 1171 | 217 |
| D | | ✓ | ✓ | 241 | 38 |
| E | —"— Vegetative Cells | | | 4779 | 1915 |
| F | | | ✓ | 1488 | 264 |
| G | | ✓ | | 1884 | 573 |
| H | | ✓ | ✓ | 2064 | 194 |
| I | *E coli.* | | | 3684 | 1547 |
| J | | | ✓ | 1448 | 371 |
| K | | ✓ | | 2365 | 1461 |
| L | | ✓ | ✓ | 835 | 302 |

**Table 2.** The number of particles remaining after a fluorescent threshold was applied for each of the non-bacterial samples collected in 2008.

| ID | Sample | Category | $n > 3\sigma$ | $n > 9\sigma$ |
|---|---|---|---|---|
| M | Bermuda grass | Fungal | 2681 | 423 |
| N | Johnson grass I | Fungal | 1209 | 259 |
| O | Johnson grass II | Fungal | 2673 | 378 |
| P | Birch pollen | Pollen | 111 | 56 |
| Q | Paper mulberry I | Pollen | 233 | 209 |
| R | Paper mulberry II | Pollen | 397 | 103 |
| S | Ragweed I | Pollen | 123 | 34 |
| T | Ragweed II | Pollen | 209 | 117 |
| U | Diesel smoke | Interferent | 11 | 5 |
| V | Grass smoke I | Interferent | 2542 | 231 |
| W | Grass smoke II | Interferent | 815 | 68 |

lected at the same time as the laboratory data, but we can see in Figure 8 that the threshold we have constructed is successful in removing the vast majority of NaCl samples collected. 55

Plots of the average fluorescent characteristics and size and shape for each sample are provided in Figures 5, 6 and 7 after a fluorescent baseline of $3\sigma$ has been applied. Similar plots have been produced using a $9\sigma$ threshold and can be found in the repository released alongside the manuscript 60 (see the code/data availability section for further details). Plots and tables for the polystyrene spheres previously published in Crawford et al. (2015) are omitted.

To provide further clarity on the variation of the samples in terms of size and fluorescence we include scatter plots of 65 each of the fluorescence channels against size for four of the

**Table 3.** The number of particles remaining after a fluorescent threshold was applied to each of the samples collected in 2014. Whether a bacterial sample was washed (w) or unwashed (unw) is specified after the sample name.

| ID | Sample | Category | $n > 3\sigma$ | $n > 9\sigma$ |
|---|---|---|---|---|
| A | *Bacillus atrophaeus* (unw) | Bacteria | 1728 | 684 |
| B | *Bacillus atrophaeus* (w) | Bacteria | 1322 | 608 |
| C | *E. coli* (unw) | Bacteria | 1290 | 632 |
| D | Puffball I | Fungal | 504 | 248 |
| E | Puffball II | Fungal | 35 | 3 |
| F | Puffball III | Fungal | 16 | 1 |
| G | Aspen Pollen | Pollen | 74 | 31 |
| H | Paper mulberry pollen | Pollen | 541 | 537 |
| I | Poplar Pollen | Pollen | 104 | 50 |
| J | Ryegrass pollen | Pollen | 21 | 15 |
| K | Fullers Earth | Interferent | 61 | 20 |
| L | NaCl | Interferent | 3 | 0 |
| M | Phosphate Buffered Saline | Interferent | 35 | 3 |

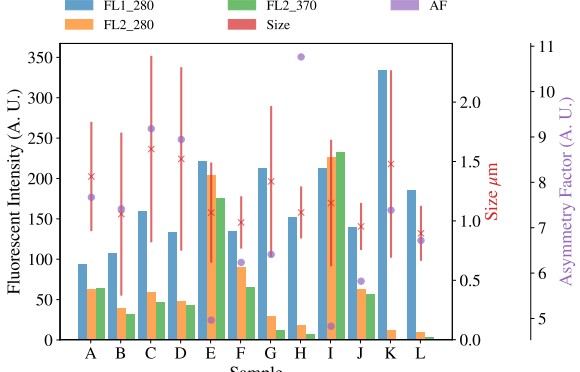

**Figure 5.** Average fluorescent characteristics for the bacterial samples collected in 2008. The error bars in red indicate a range of $\pm 1\sigma$ for each sample.

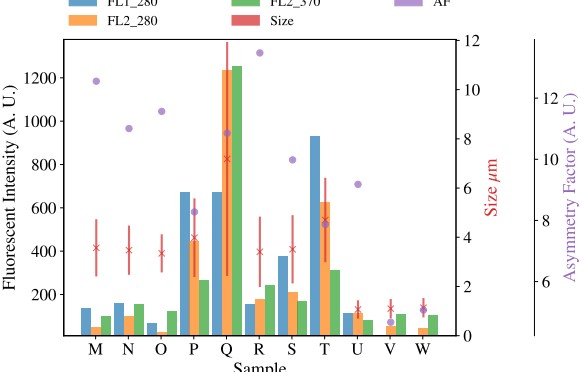

**Figure 6.** Average fluorescent characteristics for the remaining samples collected in 2008. The error bars in red indicate a range of $\pm 1\sigma$ for each sample.

samples in Figure 8. For the puffball and rye grass samples, in particular we can see that we may be measuring both fragmented and intact particles. For the interferent samples we see that a threshold of $3\sigma$ removes the vast majority of these particles. In fact the only interferent samples to measure a number of particles over a threshold of $3\sigma$ were the grass smoke samples.

The data collected is using a WIBS version 3 which is limited to a detection range of approximately $0.5\mu m$-$12\mu m$, which limits the ability of the instrument to detect intact pollen grains. The vast majority of the equivalent optical diameters (EODs) for the pollen samples collected are much lower than the measurements for intact pollen grains and are therefore likely to be pollen fragments, as was the case in Hernandez et al. (2016). The exception is the paper mulberry samples where there are differences across each of the sam-

ples. In 2008, sample Q which shows a size range similar to the other pollen samples is most likely to consist entirely of pollen fragments, whereas sample R shows a much wider size range which is likely to comprise of both fragmented and intact pollen. The collection of both fragmented and intact pollen has previously been shown to occur in Savage et al. (2017). In 2014, for sample H, the size range is much larger, consistent with the hypothesis of measuring intact pollen. Paper mulberry has been previously been sampled in (Healy et al., 2012a), using a WIBS version 4 in a low-gain mode which allows for the collection of particles up to approximately $31\mu m$. In this study, the size range of the paper mulberry was $13.6 \pm 6.2$, indicating that if sample H is intact pollen we may only be measuring part of the distribution.

It may have been possible to combine the data sets from 2008 and 2014. However, investigating if there are differ-

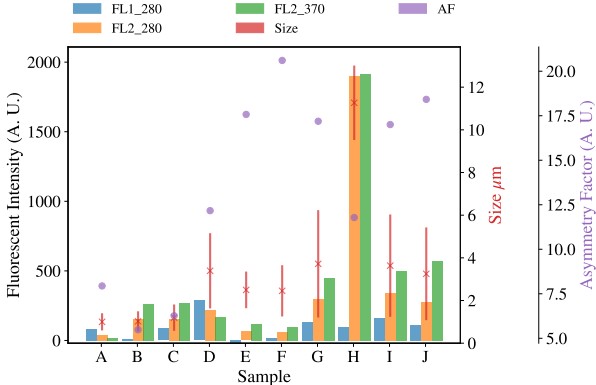

**Figure 7.** Average fluorescent characteristics for the different aerosol samples collected in 2014. The error bars in red indicate a range of $\pm 1\sigma$ for each sample.

ences in conclusions when testing different methodologies using different laboratory samples could offer insight into the reproducability of the research presented in the current study. We therefore elected to analyse the data sets separately and [5] compare and contrast the findings when testing on the PSLs and each of the data sets collected in 2008 and 2014.

## 4    Results

In Sections 4.1, 4.2, 4.3 we present the results using HAC, DBSCAN and gradient boosting respectively. A summary of [10] the findings for each method and an indication of how the number of clusters are determined are shown in Table 4.

### 4.1    Hierarchical Agglomerative Clustering

Prior to hierarchical agglomerative clustering (HAC) being applied, we labelled each particle from 1-4 to indicate [15] whether the particle was bacteria, fungal, pollen or non-biological respectively. We then considered a variety of different approaches to prepare the data which are shown in Table 5. 96 possible combinations of these considerations were applied to the data and the hierarchical agglomerative cluster-[20] ing routine was used to cluster the resultant data in each case. For each of the ninety-six hierarchies produced, the clusterings containing between 1 and 10 clusters were extracted. Subsequently, a value of the adjusted rand score comparing each of these 10 clusterings to the known labels was calcu-[25] lated. These values of the adjusted rand score would be unavailable during an ambient campaign but are used here to measure the similarity of each clustering to the known labels in order to indicate overall performance and highlight which of the first 10 clusterings was most similar to the known la-[30] bels. Values of the Calinski-Harabasz index (CH index), an index which is usually used in an ambient campaign to determine the number of clusters, were also calculated. The number of clusters in the clustering for which the maximum value

of the CH index was attained can then be compared to the clustering which is most similar to the known labels to deter-[35] mine if the CH index attains a maximum for the clustering which is most similar to the known labels.

#### 4.1.1    Impact of data preparation

Figure 9 provides an overview of the results obtained using the 96 different strategies tested. The data preparation [40] approach suggested in Crawford et al. (2015) (presented in blue) was to take logs of the size and asymmetry factor, use a size threshold of 0.8 microns, use a fluorescent threshold of 3 standard deviations above the average forced-trigger measurement, standardise using the z-score and use Ward [45] Linkage. It has also been suggested that a threshold of nine standard deviations may be more appropriate (Savage et al., 2017), so the approach suggested in Crawford et al. (2015) modified to use a threshold of $9\sigma$ is also presented (in orange)

In the case of the PSL data set, we see that HAC has pro-[50] duced a clustering with 5 clusters which is very similar to the known labels. The best performance occurred when using a fluorescent threshold of 9 standard deviations, albeit 3 standard deviations produced a similarly high value of the adjusted rand score (0.958). [55]

The maximum adjusted rand scores attained for the the laboratory generate aerosol collected in 2008 and 2014 were 0.567 and 0.747. Lower scores are to be expected since we would anticipate laboratory generated aerosol to be more complex than polystyrene latex spheres and hence more dif-[60] ficult to discriminate. The adjusted rand score of the best data strategy of the 96 tested, as indicated by the height of the green bar, is larger than the corresponding adjusted rand score for the strategy suggested in Crawford et al. (2015), indicating that potentially a different strategy may yield better [65] results. However, the best performing strategy was not consistent across both the 2008 and 2014 data.

In particular, the best strategy in 2008 was found to be: taking logs; using a size threshold of 0.8 microns; using 3 standard deviations and fluorescent threshold standardising using [70] the range and using Ward linkage. In 2014, the highest value of the adjusted rand score was obtained by not taking logs, not applying a size threshold, using a fluorescent threshold of 9 standard deviations and using the centroid linkage. Since our findings are inconsistent across the two laboratory gener-[75] ated aerosol data sets it becomes difficult to provide a better recommendation for data preparation other than the strategy suggested in Crawford et al. (2015).

In addition, there was a substantial difference between the quality of results attained when using a fluorescent threshold [80] of 3 or 9 standard deviations. In 2008, we see a decrease in the adjusted rand score from 0.482 to 0.277 when using 3 and $9\sigma$ respectively. In 2014, we see an increase in the adjusted rand score from 0.462 to 0.625 when using 3 and $9\sigma$ respectively. [85]

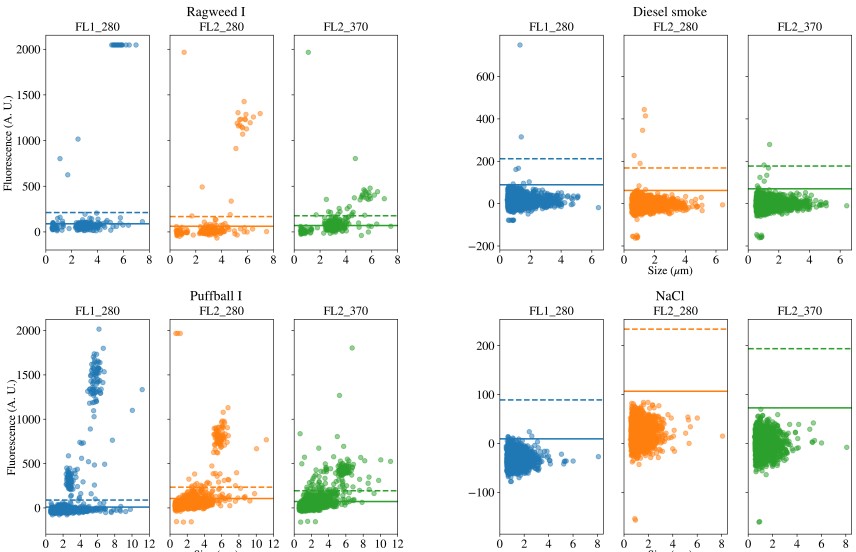

**Figure 8.** Scatter plots of fluorescence versus size for four of the samples. Two of the samples were collected in 2008 (top row) and two were collected in 2014 (bottom row); two are biological (left column) and two are non-biological (right column).

**Table 4.** Summary of findings and considerations for method selection.

| Method | Summary | # Clusters |
|---|---|---|
| HAC | – Does not rely on training data<br><br>– The conclusion we make when using the CH index may be incorrect when a large proportion of the particles are from one broad class<br><br>– How the data was prepared greatly impacted upon performance<br><br>– Particles from different categories were sometimes clustered together e.g. pollen with fungal | Determined using the maximum value of the CH index produced for clusterings between 1 and 10 clusters. |
| DBSCAN | – Produced a clustering which contained three distinct clusters each containing primarily one broad class of bioaerosol in the case of one of the data sets<br><br>– Data preparation greatly impacted upon performance<br><br>– It is not clear at this point whether the values of epsilon and the minimum number of points would be applicable to ambient data | Naturally determined by setting epsilon and the minimum number of points required for a neighbourhood. |
| Gradient Boosting | – Performance was consistently good irregardless of data preparation provided that a threshold, either 3 or 9 standard deviations, was applied to the fluorescence<br><br>– Relies on adequate training data being collected and it is not clear at this point whether the data collected will be sufficient. | Always the same as the number of groups in the training data |

**Table 5.** Outline of the different approaches tested when using Hierarchical Agglomerative Clustering

| Consideration | Option |
|---|---|
| Take Logs | True or False |
| Size Threshold | None or 0.8 |
| Fluorescent Threshold | None, $3\sigma$ or $9\sigma$ |
| Standardisation | Z-score or Range |
| Linkage | Ward, Centroid, Median or Single |

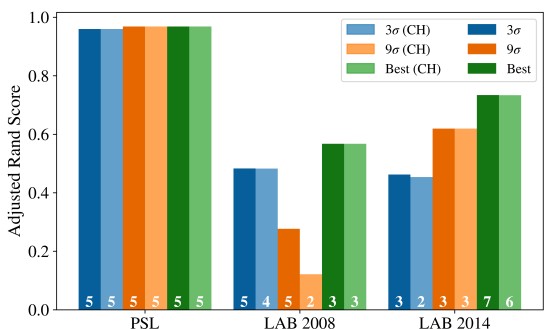

**Figure 9.** Performance of Hierarchical Agglomerative Clustering using the adjusted rand score for the data sets tested across different data preparation strategies. The number of clusters concluded in each case is indicated at the bottom of each bar.

It is possible that the difference in performance when using the different thresholds could be in part explained by the fluorescent threshold in 2014 being constructed using forced trigger data collected at a different time to the laboratory data, or by the fluorescence properties differing across the two data sets. But this differing behaviour when using different data preparation does need to be investigated further with additional laboratory data sets and in the context of ambient data. Nonetheless, the differing conclusions across the two data sets as to which data preparation is preferable does highlight the importance of repeating data collection and demonstrating conclusions are consistent across multiple experiments.

The adjusted rand score is often quite difficult to interpret, so we provide matching matrices for the best and worst case scenario using the current data preparation strategy in Tables 6 and 7. In the best case scenario we are able to discriminate between the pollen and the rest of the data placing $86.8\%$ of the pollen into Cluster 2. Most of the bacteria is also placed into Cluster 3 with $66.6\%$ of the fungal spores. A third of the fungal spores are differentiated from the rest of the data and placed into Cluster 1. In the worst case scenario two clusters are provided both primarily containing bacteria. In this case we can conclude that algorithm has failed to differentiate between any of the biological classes, in part due to the CH index concluding there are 2 clusters.

### 4.1.2    Impact of the Calinski-Harabasz Index

At the base of each bar in Figure 9 we provided the number of clusters in the clustering for which the adjusted rand score presented was obtained. For the darker bars, this number represents the number of clusters in the clustering for which the highest value of the adjusted rand score was obtained across the clusterings containing between 1 and 10 clusters. For the lighter bars, this number represents the number of the clusters in the clustering for which a maximum of the CH index was attained.

There are three different scenarios that occur. First, the Calinski-Harabasz index attains a maximum for the clustering which is most similar to the known labels e.g. for the PSL data using a fluorescent threshold of 3 standard deviations, the clustering which is most similar to the known labels (shown in the darker bar) contains 5 clusters which is the same as the clustering for which a maximum of the CH index is attained (shown in the lighter bar). Second, the Calinski-Harabasz index attains a maximum for a different clustering which is most similar to the known labels, but the conclusion does not have a large impact on performance. For example, in 2008 using a fluorescent threshold of 3 standard deviations, the clustering which is most similar to the known labels contains 5 clusters, whereas the clustering for which the CH index attains a maximum contains 4 clusters. However, the heights of bars are nearly the same. In this case, a very small cluster has been merged in the hierarchy from 5 to 4 clusters resulting in the 4 and 5 cluster clusterings being extremely similar and consequently the fact that the CH index has attained a maximum at 4 clusters instead of 5 is not concerning, since concluding 4 clusters instead of 5 has very little impact upon performance.

The final case is in 2008, using a fluorescent threshold of 9 standard deviations. Here the clustering which is most similar to the known labels is the clustering containing 5 clusters, whereas the CH index attains a maximum for the clustering containing only 2 clusters. The 2 cluster solution in this case is very dissimilar from the known labels.

In the cases where a maximum for the CH index was attained for a clustering containing 2 clusters i.e. in 2008 using $9\sigma$ and in 2014 using $3\sigma$, $78.6\%$ and $76.5\%$ of the particles were from a bacterial sample. Conversely in 2008 using $3\sigma$ and in 2014 using $9\sigma$, $65.4\%$ and $68.4\%$ of the particles analysed were bacteria.

To investigate the possibility of a relationship between the proportion of the data which is contained in the category containing the largest number of particles and the tendency of the CH index to conclude that there are 2 clusters we produced data simulated from 3 normal distributions in 3 dimensions. Each of the clusters was centred around [0, 0, 0], [5, 5, 5], [10, 10, 10] and the co-variance matrix was set to

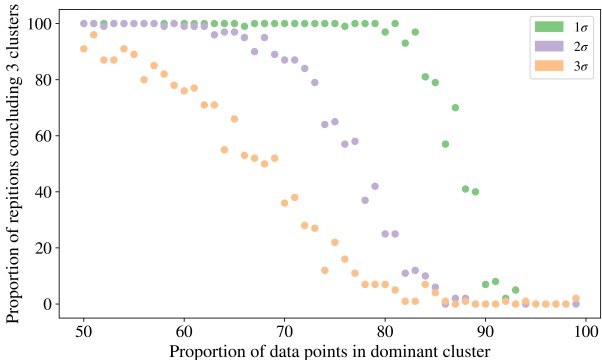

**Figure 10.** Percentage of simulations for which the CH index attained a maximum for the clustering containing 3 clusters against the proportion of the data which is placed into a dominant cluster.

$\sigma I_3$, where $I_3$ is the 3 by 3 identity matrix. The value of $\sigma$ was varied from 1-3 to produce a range of variation in the simulations. We elected to produce this simulated data from normal distributions rather than the laboratory data collected
to remove any potential confounding issues such as the fluorescent threshold used. The proportion of the data that was contained in the dominant cluster was varied from 50% to 99%. Each simulation was repeated 100 times to provide an indication of the frequency the CH index attains a maximum
for the 3 cluster solution.

In Figure 10 we see that there is a point where the frequency for which the CH index attains a maximum for the clustering contains 3 clusters starts to decrease. The proportion of data points that needs to be placed in the dominant
cluster before this decrease in performance of the CH index is seen decreases as the variability in the data increases.

This incorrect conclusion when using the CH index when analysing data for which a large proportion of data is of one particular type is problematic when analysing biological
aerosol, since we may expect the quantity of bacteria to be an order of magnitude greater than than the fungal spores, and for the quantity of fungal spores to be an order of magnitude greater than the pollen (Després et al., 2012; Gabey, 2011). In future studies it may therefore be necessary to explore the
use of other indices for determining the number of clusters.

### 4.1.3 Breakdown of the hierarchies

To more clearly understand how data has been clustered using HAC we have presented dendrograms for the laboratory data collected in 2008 and 2014 in Figures 11 and 12 along-
30 side heat maps of the matching matrices to indicate the cluster composition of the 10 cluster solution broken down by sample. The hierarchy produced using the strategy suggested in Crawford et al. (2015) is presented at the top of each plot whereas a modification of this strategy using a threshold of

**Table 6.** Matching matrix for the best case scenario when using the current data preparation strategy with $9\sigma$ on the data collected in 2014

|      | bacteria | fungal spores | pollen | non-biological |
|------|----------|---------------|--------|----------------|
| CL1  | 4        | 80            | 13     | 5              |
| CL2  | 85       | 4             | 550    | 3              |
| CL3  | 1835     | 168           | 70     | 15             |

9 standard deviations as suggested by Savage et al. (2017) is 35 presented at the bottom.

Each row of the heat map corresponds to a particular cluster and each column corresponds to a particular sample. The intensity of each box corresponds to the quantity of particles placed into a particular cluster from a particular sam- 40 ple. Bacterial, fungal, pollen and non-biological samples are grouped together in blue, green, orange and black respectively. Different scales are used for the different groups to prevent the dominant class from obscuring information in the other classes. 45

In 2008, the majority of the Bacteria is placed into a single cluster for both $3\sigma$ and $9\sigma$. The fungal and a number of pollen particles are placed into the same two clusters when using $3\sigma$ and into one cluster when using $9\sigma$. The non-biological samples, consisting primarily of grass smoke, are clustered 50 mostly with bacterial samples, possibly due to their similar size. In addition, there two clusters when using $3\sigma$ and three clusters when using $9\sigma$ containing primarily pollen.

In 2014, pollen has been placed primarily into 1 or 2 clusters. Some of the fungal samples have been placed into a 55 singleton cluster. For both thresholds the bacteria is grouped with some of the fungal samples. The non-biological material has almost entirely been removed by the threshold and the remaining material has been divided among a number of the clusters. 60

In both 2008 and 2014, some of the material has been segregated into clusters containing primarily one broad class of biological aerosol. However, a number of fungal particles has been grouped with pollen samples in the case of 2008 and a number of the fungal samples have been grouped with bac- 65 terial particles in 2014. The more successful segregation of pollen in 2014 may be due to the much larger size range for the paper mulberry sample, whereas in 2008 the fungal and pollen material may be grouped due to presence of a larger number of pollen fragments. It is therefore important when 70 interpreting results from an ambient campaign that it is possible that clusters may contain more than one broad biological class.

Also note that this potentially undesirable grouping of material from two different classes has occurred prior to the final 75 stages of the algorithm and therefore will be apparent in the final solution regardless of the number of clusters concluded, and cannot be rectified by using a different validation index.

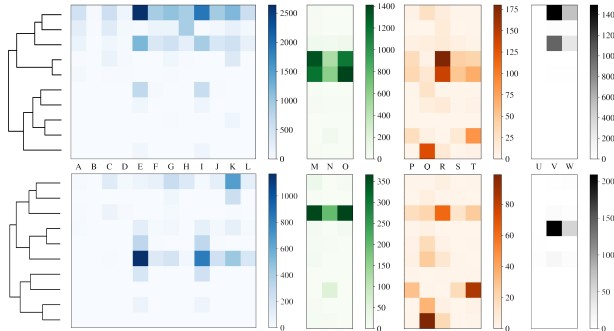

**Figure 11.** Dendrogram truncated at 10 clusters (left) for laboratory data collected in 2008 alongside heat map of matching matrix (right) indicating cluster composition by each sample segregated by bacteria, fungal, pollen and non-biological in blue, green, orange and black respectively. Separate scales are used for each broad class to prevent dominant class obscuring detail in the other classes. Hierarchies for $3\sigma$ (top) and $9\sigma$ (bottom) are presented.

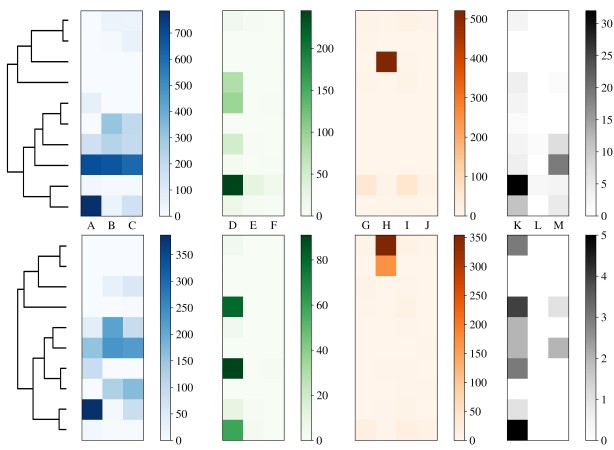

**Figure 12.** Dendrogram trunctated at 10 clusters (left) for laboratory data collected in 2014 alongside heat map of matching matrix (right) indicating cluster composition by each sample segregated by bacteria, fungal, pollen and non-biological in blue, green orange and black respectively. Separate scales are used for each broad class to prevent the dominant class obscuring detail in the other classes. Hierarchies for $3\sigma$ and $9\sigma$ (bottom) are presented.

## 4.2 DBSCAN

One of the main difficulties of using DBSCAN is selecting the minimum number of points to form a neighbourhood and the radius of the neighbourhood (Khan et al., 2014). For $3\sigma$
5 and $9\sigma$ using z-score standardisation, taking logs of the size and asymmetry factor and removing particles smaller than $0.8$ microns we repeat the DBSCAN algorithm for a variety

**Table 7.** Matching matrix for the worst case scenario when using the current data preparation strategy with $9\sigma$ on the data collected in 2008

|  | bacteria | fungal spores | pollen | non-biological |
|---|---|---|---|---|
| CL1 | 547 | 69 | 298 | 0 |
| CL2 | 6373 | 991 | 221 | 304 |

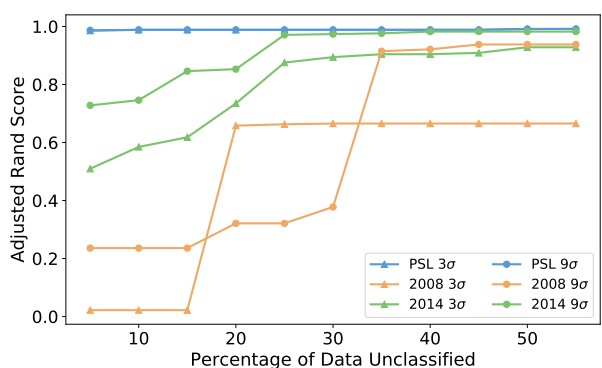

**Figure 13.** Adjusted rand score using different thresholds of percentage of points we allow to be left in the analysis for DBSCAN.

of $\epsilon$ (neighbourhood radii) and minimum number of points values. The range of values of $\epsilon$ we test is $0.1, 0.2, \cdots, 1.0$. The range of minimum number of points is set using the 10 following range relative to the number of particles collected $0.1\%, 0.2\%, \cdots, 1.0\%, 2.0\%, \cdots, 10.0\%$.

We found wide variety of performance across the different parameters. Often high accuracy could be obtained when using a high value of the minimum number of points but this 15 resulted in removing a substantial portion of the data. In Figure 13 we filter our results using a range of thresholds for the maximum number of points that can be left unclassified $(5\%, 10\%, \cdots 60\%)$ and plot the corresponding best performance under this filter. In all the data sets there was a point 20 of diminishing returns where no further benefit could be attained by removing any more of the data. In the case of the PSL data, this point happened after removing around $5\%$ of the particles. For the laboratory data sets between $25$ and $40\%$ of the data was left unclassified before a peak in per- 25 formance was attained. Nonetheless, we note in the case of the laboratory data collected in $2014$ and using a $9\sigma$ fluorescent threshold, we can attain performance similar to that which we attain for the PSL data.

In order to investigate further a choice of $\epsilon$ and the mini- 30 mum number of points which would maximise performance in terms of the adjusted rand score we plot the adjusted rand score for each test across all of the data sets. In Figure 14 we see that there is a large window of different values for which a higher value of the adjusted rand score can be achieved on 35 the PSLs. Contrary to this, in 2008 when using $9\sigma$ there is a

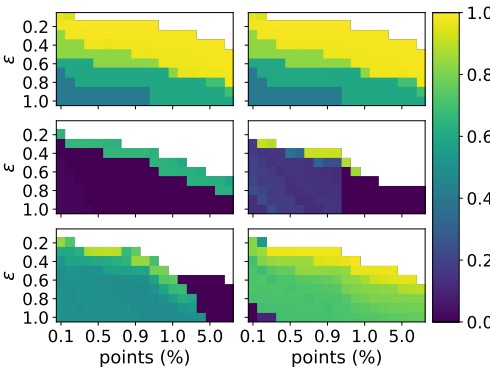

**Figure 14.** Adjusted rand score for DBSCAN, over a range different values of $\epsilon$ and minimum number of points required to form a neighborhood. The minimum number of points is expressed relative to the total number of points. The columns correspond to 3 and $9\sigma$ respectively. The rows correspond to the PSL, 2008 and 2014 data respectively.

very narrow window for which higher values of the adjusted rand score could be attained. It can also be seen that as $\epsilon$ increases the number of points required to create a cluster needs to be increased to compensate.

Overall our results indicate setting $\epsilon = 0.3$ and $\epsilon = 0.4$ when using $3\sigma$ and $9\sigma$ respectively. The best results can then be obtained by setting the number of points between $0.4\%$ and $0.7\%$ of the data when using an $\epsilon$ of 0.3 and $0.7\%$ and $1.0\%$ when using an $\epsilon$ of 0.4. However, future research will be required to demonstrate these conclusions are applicable when studying ambient data.

We provide matching matrices for the worst and best case scenarios in Tables 8 and 9. We see that in the best case scenario, leaving a decent proportion of data left unclassified we are able to produce three distinct clusters containing predominantly one broad class of biological aerosol. In the worst case scenario we manage only to distinguish between the bacteria from the fungal spores combined with the pollen.

In the worst case scenario i.e. using $3\sigma$, on the 2008 data we fail to remove a sizeable fraction of the non-biological particles, which was also the case when using HAC, however we would have expected that the algorithm would leave the particles unclassified. There is some argument that this worst case scenario could be circumvented by simply using the $9\sigma$ threshold instead. But further research needs to be conducted on the handling of non-biological material that appears fluorescent in the instrument.

### 4.3 Gradient Boosting

We conducted a similar analysis varying data preparation approaches as in Section 4.1. We found data preparation to have a very small impact upon performance when using Gradient Boosting as long as some kind of fluorescence threshold is

**Table 8.** Matching matrix for the best case scenario when using DBSCAN with $9\sigma$, $\epsilon = 0.4$ and a minimum number of points of $0.7\%$ on 2014 data.

|  | bacteria | fungal | pollen | non-biological |
|---|---|---|---|---|
| Unclassified | 329 | 169 | 134 | 16 |
| CL1 | 0 | 0 | 490 | 0 |
| CL2 | 12 | 80 | 4 | 0 |
| CL3 | 1583 | 3 | 5 | 7 |

**Table 9.** Matching matrix for the worst case scenario when using DBSCAN with $3\sigma$, $\epsilon = 0.3$ and a minimum number of points of $0.4\%$ on 2008 data

|  | bacteria | fungal | pollen | non-biological |
|---|---|---|---|---|
| Unclassified | 5858 | 1893 | 636 | 752 |
| CL1 | 15025 | 15 | 44 | 2616 |
| CL2 | 80 | 4655 | 393 | 0 |

applied where a high value of the adjusted rand score was obtained regardless of whether we took logs, what standardisation was used or the size threshold imposed.

Figure 15 shows the performance using $3\sigma$ and $9\sigma$ using z-score, taking logs and applying a size threshold of 0.8 microns. High performance was attained across both laboratory generated aerosol data sets and for the PSLs. As we did in the previous sections we provide matching matrices of the worst-case scenario and best case scenario when using gradient boosting using the current data preparation in Tables 10 and 11. In the best case scenario we provide a very good classification with very small errors (AR=0.933).

In the worst case scenario a similar performance is achieved (AR = 0.882). Nonetheless, a few particles are incorrectly classified within the fungal spore and pollen classes. The classification for the bacteria is still very strong and most of the remaining non-biological particles are correctly classified. The non-biological samples have been removed from this data set prior to gradient boosting being applied when using a fluorescent threshold of either $3\sigma$ or $9\sigma$. We elect to remove these particles since too few of the non-biological samples that exceed either threshold to produce a viable training class.

### 4.4 K-means

Similar to the findings presented in Ruske et al. (2017), k-means performed poorly and hence the results are omitted from the main text. The results are available in the repository published alongside the manuscript (see the code/data availability section for further details).

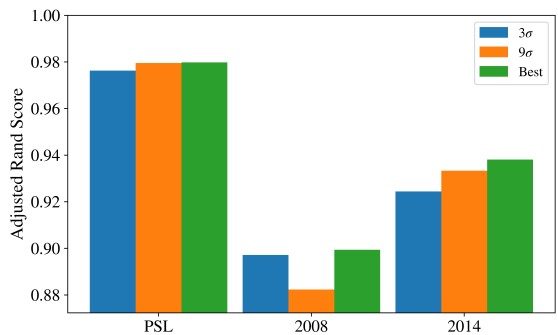

**Figure 15.** Performance of Gradient Boosting for the different data sets when using $3\sigma$ and $9\sigma$.

**Table 10.** Matching matrix for the best case scenario when using Gradient Boosting. This is when using $9\sigma$ on 2014 data.

|  | bacteria | fungal | pollen |
|---|---|---|---|
| bacteria | 1911 | 8 | 19 |
| fungal | 7 | 219 | 29 |
| pollen | 6 | 25 | 595 |

## 5   Conclusions

We evaluated a variety of different methods that could be used for classification of biological aerosol. Gradient Boosting offered the best performance consistently across the different data preparation strategies and the different data sets tested. That being said it is unclear at this point how this will translate to ambient data and whether or not the training data currently collected will be sufficient to outline the variety of environments that could potentially be studied.

Should there not be sufficient training data available an unsupervised approach may be required. In this case, a possible alternative to HAC is provided. In the best case scenario DBSCAN, despite leaving a decent proportion of the data unclassified, was able to produce three distinct clusters containing predominantly one biological class each.

To the best of our knowledge this is the first manuscript using DBSCAN to classify biological aerosol using the WIBS. So we will need to continue to evaluate the performance of this algorithm in the context of the ambient setting. In particular, we have provided details of what we believe to be sensible selections of epsilon and the minimum number of points on the basic of the laboratory data collected. However, it is unclear at this point how effective these selections will be when analysing ambient data.

When applied the laboratory generated aerosol tested, we found that performance of HAC was in general much lower than what was achieved previously using the PSLs (Craw-

**Table 11.** Matching matrix for the worst case scenario when using Gradient Boosting. This is when using $9\sigma$ on 2008 data.

|  | bacteria | fungal | pollen | non-biological |
|---|---|---|---|---|
| bacteria | 6852 | 85 | 76 | 8 |
| fungal | 56 | 898 | 147 | 2 |
| pollen | 8 | 72 | 293 | 0 |
| non-biological | 4 | 5 | 3 | 294 |

ford et al., 2015). Performance was heavily dependent on the data preparation strategy used, and often results could vary substantially between different strategies and data sets, potentially due to differences in the fluorescence measurements across the two data sets. A potential issue with the CH index is highlighted, whereby we see a failure of the index to determine the correct number of clusters as the size of the dominant class and variation in the data increases. Some of the pollen samples were clustered with the fungal samples when analysing the data from 2008. A number of the pollen particles may be fragmented which may explain why this grouping may occur. Similarly grass smoke was grouped with the bacterial samples, potentially due to their similar size. Caution will therefore be required when applying the HAC algorithm to ambient data, and it must be noted in particular that material from two different classes may be placed into the same cluster and that the CH index may indicate an incorrect number of clusters if the data collected contains a significant quantity of one particular type of particle.

In the future, more laboratory generated aerosol particles will need to be collected to continue to evaluate the performance of the algorithms which we use. In addition, when Gradient Boosting was used we failed to classify the some of the pollen and fungal spore samples analysed. It is therefore possible that higher spectral instruments such as the spectral intensity bioaerosol sensor (Nasir et al., 2018), will be required to provide a more accurate classification.

*Code and data availability.* Part of the code used produce the above manuscript is part of an ongoing development of a software suite for analysis of various UV-LIF instruments, are available at https://github.com/simonruske/UVLIF upon publication. Other code not currently included within the software package i.e. code files which are used to produce the plots and figures specific to the current manuscript are available at https://github.com/simonruske/AMT-2018-126.

The data used is available upon request by contacting the lead author.

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

## Appendix A:  Comparison of particle size with other studies

To contextualise the samples collected in the current study we examined the literature to find similar studies using the WIBS as well as other studies using microscopy. In the case of most of the samples we were able to find a paper on the same or similar species of particle which are presented in Table A1.

## Appendix B:  ABC counts / Average particle sizes

To aid in comparing the data presented with other studies, we have presented Tables B1 and B2 which is very similar to the table in the appendices of Hernandez et al. (2016). The same information but using a $9\sigma$ threshold instead is presented in Table B3.

## Appendix C:  Summary of average properties of the different data sets

In the following section we summarise mean and standard deviations in each of the five measurements in each of the samples collected in 2008 and 2014. The properties presented in Tables C1, C2, C3 and C4 are after a size threshold of $0.8\mu$m is imposed and a fluorescent threshold of either $3\sigma$ or $9\sigma$ has been applied. These summary statistics presented are prior to any log-transformations or data standardisation has been applied.

*Competing interests.*  There are no competing interests that the authors are aware of

*Acknowledgements.* Simon Ruske is funded by NERC (NERC Grant number: NE/L002469/1) and the University of Manchester.

**Table A1.** Average particle sizes for the current study compared with other studies. The sizes presented here are collated from the following studies [1] Healy et al. (2012a), [2] Savage et al. (2017), [3] Hernandez et al. (2016), [4] Pierucci (1978), [5] Carrera et al. (2007), [6] Crotzer and Levetin (1996), [7] Geiser et al. (2000), [8] Pinnick et al. (1995), [9] Fumanal et al. (2007), [10] Mäkelä (1996), [11] Kang et al. (2007), [2008] & [2014] the current study.

| Sample | Measurement type | Size ($\mu$m) | Reference |
|---|---|---|---|
| Paper mulberry | WIBS4 low-gain | $13.6 \pm 6.2$ | [1] |
| | WIBS3 | $7.18 \pm 4.74$ | [2008] |
| | WIBS3 | $3.41 \pm 1.43$ | [2008] |
| | WIBS3 | $11.27 \pm 1.74$ | [2014] |
| | Miscroscopy | $13.8$ | [11] |
| Ragweed pollen | WIBS4 low-gain | $24.5 \pm 7.6$ | [1] |
| | WIBS3 | $3.51 \pm 1.38$ | [2008] |
| | WIBS3 | $4.70 \pm 1.71$ | [2008] |
| | Microscopy | $13.02 \pm 0.12 - 14.86 \pm 0.16$ | [9] |
| Birch pollen | WIBS4 low-gain | $19.0 \pm 9.2$ | [1] |
| Betula lenta, nigra & populifolia | WIBS4 | $2.5 \pm 4.2$ | [3] |
| Birch pollen | WIBS3 | $3.98 \pm 1.59$ | [2008] |
| Betula (various) | Microscopy | $17.31 \pm 0.08 - 24.36 \pm 1.59$ | [10] |
| White poplar | WIBS4A | $18.7 \pm 1.9$ | [2] |
| White poplar fragments | WIBS4A | $7.4 \pm 4.0$ | [2] |
| Aspen pollen | WIBS3 | $3.72 \pm 2.49$ | [2014] |
| Poplar pollen | WIBS3 | $3.63 \pm 2.39$ | [2014] |
| Bermuda grass smut | WIBS4 high-gain | $4.7 \pm 2.2$ | [1] |
| | WIBS3 | $3.57 \pm 1.16$ | [2008] |
| | Microscopy | $6.7 \times 6.5$ | [6] |
| Johnson grass smut | WIBS4 high-gain | $8.9 \pm 1.5$ | [1] |
| | WIBS3 | $3.47 \pm 1.00$ | [2008] |
| | WIBS3 | $3.35 \pm 0.78$ | [2008] |
| | Microscopy | $13.9 \times 12.6$ | [6] |
| Puffball spores | Microscopy | $3.5 \pm 0.24$ | [7] |
| | WIBS3 | $2.50 \pm 0.85$ | [2008] |
| | WIBS3 | $2.45 \pm 1.16$ | [2008] |
| | WIBS3 | $3.39 \pm 1.76$ | [2008] |
| | Fluorescence particle counter | $2$-$4$ | [8] |
| *Bacillus atrophaeus* spores | WIBS4A | $2.2 \pm 0.4$ | [2] |
| | WIBS3 | $1.00 \pm 0.40 - 1.60 \pm 0.78$ | [2008, 2014] |
| | Microscopy | $1.22 \pm 0.12$ (length) | |
| | | $0.65 \pm 0.05$ (diameter) | [5] |
| *Bacillus atrophaeus* vegetative cells | WIBS3 | $1.06 \pm 0.68 - 1.60 \pm 0.78$ | [2008] |
| *E. coli* | WIBS4A | $1.2 \pm 0.3$ | [2] |
| | WIBS4 | $0.9 \pm 0.4$ | [3] |
| | WIBS3 | $0.89 \pm 0.23 - 1.48 \pm 0.79$ | [2008, 2014] |
| | Microscopy | $1.67 - 3.08$ (length) | |
| | | $0.69 - 0.84$ (diameter) | [4] |

**Table B1.** For the data collected in 2008, a summary of size and fluorescent measurements for each sample to include: the number of particles in the sample (total), average equivalent optical diameter (EOD), standard deviation of the size ($\sigma$), the number of points that exceeded a fluorescent threshold of 3 standard deviations above the average forced trigger measurement ($n > 3\sigma$), and ABC counts using a $3\sigma$ threshold.

| | n | EOD | $\sigma$ | $n > 3\sigma$ | A | B | AB | C | AC | BC | ABC |
|---|---|---|---|---|---|---|---|---|---|---|---|
| **Bacteria** | | | | | | | | | | | |
| *Bacillus atrophaeus* (unwashed) | 5778 | 1.4 | 0.5 | 1015 | 322 | 200 | 74 | 113 | 48 | 90 | 168 |
| —"— (unwashed, diluted) | 1525 | 1.1 | 0.7 | 82 | 65 | 6 | 3 | 4 | 0 | 3 | 1 |
| —"— (washed) | 4694 | 1.6 | 0.8 | 1246 | 728 | 107 | 191 | 18 | 29 | 5 | 168 |
| —"— (washed, diluted) | 1786 | 1.5 | 0.8 | 280 | 183 | 21 | 30 | 9 | 10 | 12 | 15 |
| —"— vegetative cells (unwashed) | 6142 | 1.1 | 0.4 | 5546 | 409 | 693 | 771 | 75 | 79 | 287 | 3232 |
| —"— vegetative cells (unwashed, diluted) | 2192 | 1 | 0.2 | 1739 | 484 | 279 | 326 | 30 | 26 | 67 | 527 |
| —"— vegetative cells (washed) | 6002 | 1.3 | 0.6 | 1961 | 1797 | 3 | 139 | 0 | 1 | 1 | 20 |
| —"— vegetative cells (washed, diluted) | 2827 | 1.1 | 0.2 | 2218 | 2178 | 3 | 36 | 0 | 0 | 1 | 0 |
| *E. coli* (unwashed) | 4956 | 1.2 | 0.5 | 4097 | 366 | 578 | 174 | 179 | 69 | 868 | 1863 |
| —"— (unwashed, diluted) | 2508 | 1 | 0.2 | 1778 | 751 | 309 | 82 | 99 | 27 | 263 | 247 |
| —"— (washed) | 5669 | 1.5 | 0.8 | 2627 | 2508 | 1 | 99 | 0 | 0 | 0 | 19 |
| —"— (washed, diluted) | 2104 | 0.9 | 0.2 | 1390 | 1383 | 0 | 5 | 0 | 0 | 2 | 0 |
| **Fungal** | | | | | | | | | | | |
| Bermuda grass smut | 5220 | 3.6 | 1.2 | 2681 | 1446 | 7 | 34 | 271 | 495 | 81 | 347 |
| Johnson grass smut I | 2157 | 3.5 | 1 | 1211 | 76 | 3 | 3 | 796 | 128 | 92 | 113 |
| Johnson grass smut II | 5091 | 3.3 | 0.8 | 2675 | 217 | 8 | 1 | 1939 | 270 | 132 | 108 |
| **Pollen** | | | | | | | | | | | |
| Birch pollen | 164 | 4 | 1.6 | 112 | 16 | 1 | 0 | 29 | 7 | 8 | 51 |
| Paper mulberry pollen I | 295 | 7.2 | 4.7 | 237 | 16 | 2 | 9 | 2 | 0 | 21 | 187 |
| Paper mulberry pollen II | 735 | 3.4 | 1.4 | 405 | 159 | 2 | 9 | 72 | 59 | 37 | 67 |
| Ragweed pollen I | 241 | 3.5 | 1.4 | 127 | 24 | 1 | 0 | 57 | 12 | 7 | 26 |
| Ragweed pollen II | 328 | 4.7 | 1.7 | 209 | 21 | 0 | 1 | 41 | 16 | 15 | 115 |
| **Non-biological** | | | | | | | | | | | |
| Diesel smoke | 7900 | 1.1 | 0.4 | 16 | 3 | 4 | 0 | 5 | 0 | 0 | 4 |
| Grass smoke I | 9212 | 1.1 | 0.4 | 2976 | 1 | 234 | 0 | 2004 | 0 | 737 | 0 |
| Grass smoke II | 5245 | 1.1 | 0.4 | 900 | 3 | 51 | 0 | 668 | 0 | 176 | 2 |

**Table B2.** For the data collected in 2014, a summary of size and fluorescent measurements for each sample to include: the number of particles in the sample (total), average equivalent optical diameter (EOD), standard deviation of the size ($\sigma$), the number of points that exceeded a fluorescent threshold of 3 standard deviations above the average forced trigger measurement ($n > 3\sigma$), and ABC counts using a $3\sigma$ threshold.

| | n | EOD | $\sigma$ | $n > 3\sigma$ | A | B | AB | C | AC | BC | ABC |
|---|---|---|---|---|---|---|---|---|---|---|---|
| **Bacteria** | | | | | | | | | | | |
| *Bacillus atrophaeus* (washed) | 3321 | 1 | 0.4 | 2685 | 2545 | 1 | 15 | 1 | 81 | 1 | 41 |
| *Bacillus atrophaeus* (unwashed) | 2896 | 1 | 0.5 | 2248 | 85 | 15 | 2 | 1166 | 88 | 350 | 542 |
| *E. coli* (unwashed) | 2534 | 1.2 | 0.6 | 1640 | 268 | 10 | 5 | 439 | 239 | 48 | 631 |
| **Fungal** | | | | | | | | | | | |
| Puffball spores I | 1739 | 2.5 | 0.8 | 35 | 3 | 1 | 0 | 27 | 1 | 3 | 0 |
| Puffball spores II | 553 | 2.5 | 1.2 | 16 | 2 | 0 | 0 | 12 | 1 | 0 | 1 |
| Puffball spores III | 1627 | 3.4 | 1.8 | 506 | 79 | 4 | 73 | 168 | 7 | 68 | 107 |
| **Pollen** | | | | | | | | | | | |
| Aspen pollen | 398 | 3.7 | 2.5 | 74 | 5 | 1 | 0 | 35 | 1 | 11 | 21 |
| Poplar pollen | 375 | 3.6 | 2.4 | 104 | 7 | 0 | 3 | 45 | 4 | 21 | 24 |
| Paper mulberry pollen I | 565 | 11.3 | 1.7 | 543 | 3 | 0 | 1 | 4 | 0 | 35 | 500 |
| Ryegrass pollen | 47 | 3.3 | 2.1 | 21 | 0 | 0 | 0 | 6 | 0 | 7 | 8 |
| **Non-biological** | | | | | | | | | | | |
| Fullers' earth | 3064 | 3.6 | 2.8 | 62 | 40 | 1 | 0 | 8 | 4 | 3 | 6 |
| Phosphate buffered saline | 3226 | 1.2 | 1.6 | 50 | 29 | 7 | 0 | 11 | 1 | 0 | 2 |
| NaCl | 2197 | 1.4 | 0.8 | 6 | 6 | 0 | 0 | 0 | 0 | 0 | 0 |

**Table B3.** Summary of properties for samples collected from 2008 and 2014 respectively using a fluorescent threshold of $9\sigma$.

| | EOD | $\sigma$ | $n > 9\sigma$ | A | B | AB | C | AC | BC | ABC |
|---|---|---|---|---|---|---|---|---|---|---|
| **2008** | | | | | | | | | | |
| **Bacteria** | | | | | | | | | | |
| *Bacillus atrophaeus* (unwashed) | 2.2 | 0.6 | 34 | 9 | 2 | 3 | 13 | 1 | 3 | 3 |
| —"— (unwashed, diluted) | 2.7 | 1.8 | 4 | 1 | 0 | 0 | 0 | 0 | 2 | 1 |
| —"— (washed) | 2.6 | 1.1 | 217 | 182 | 0 | 9 | 0 | 5 | 0 | 21 |
| —"—(washed, diluted) | 2.6 | 1 | 38 | 28 | 0 | 1 | 1 | 0 | 6 | 2 |
| —"— vegetative cells (unwashed) | 1.3 | 0.6 | 2051 | 273 | 326 | 132 | 33 | 19 | 184 | 1084 |
| —"— vegetative cells (unwashed, diluted) | 1.2 | 0.3 | 278 | 121 | 72 | 24 | 2 | 2 | 14 | 43 |
| —"— vegetative cells (washed) | 1.7 | 0.9 | 581 | 567 | 0 | 13 | 0 | 0 | 0 | 1 |
| —"— vegetative cells (washed, diluted) | 1.3 | 0.3 | 196 | 196 | 0 | 0 | 0 | 0 | 0 | 0 |
| *E.coli* (unwashed) | 1.5 | 0.7 | 1676 | 343 | 97 | 23 | 92 | 30 | 334 | 757 |
| —"— (unwashed, diluted) | 1.1 | 0.2 | 413 | 333 | 23 | 4 | 12 | 4 | 17 | 20 |
| —"— (washed) | 1.7 | 0.9 | 1516 | 1506 | 2 | 7 | 0 | 0 | 0 | 1 |
| —"— (washed, diluted) | 1.1 | 0.3 | 349 | 348 | 0 | 0 | 0 | 0 | 1 | 0 |
| **Fungal** | | | | | | | | | | |
| Bermuda grass smut | 4 | 1.5 | 423 | 118 | 10 | 14 | 133 | 19 | 37 | 92 |
| Johnson grass smut | 4.2 | 1.3 | 259 | 0 | 0 | 1 | 171 | 0 | 29 | 58 |
| Johnson grass smut II | 3.8 | 1 | 378 | 2 | 2 | 0 | 340 | 0 | 29 | 5 |
| **Pollen** | | | | | | | | | | |
| Birch pollen | 4.5 | 1.8 | 57 | 7 | 0 | 0 | 9 | 0 | 2 | 39 |
| Paper mulberry | 7.8 | 4.6 | 212 | 22 | 0 | 7 | 3 | 0 | 30 | 150 |
| Paper mulberry II | 3.9 | 2.1 | 107 | 17 | 2 | 2 | 21 | 0 | 39 | 26 |
| Ragweed pollen | 4.7 | 1.4 | 34 | 0 | 0 | 0 | 11 | 0 | 2 | 21 |
| Ragweed pollen II | 5.5 | 1.5 | 117 | 2 | 0 | 0 | 9 | 0 | 10 | 96 |
| **Non-biological** | | | | | | | | | | |
| Diesel smoke | 1.1 | 0.2 | 6 | 0 | 3 | 1 | 1 | 0 | 0 | 1 |
| Grass smoke | 2 | 0.5 | 236 | 0 | 1 | 0 | 218 | 0 | 17 | 0 |
| Grass smoke | 2 | 0.4 | 68 | 0 | 0 | 0 | 64 | 0 | 4 | 0 |
| **2014** | | | | | | | | | | |
| **Bacteria** | | | | | | | | | | |
| *Bacillus atrophaeus*(washed) | 1.3 | 0.5 | 735 | 721 | 0 | 0 | 0 | 12 | 1 | 1 |
| *Bacillus atrophaeus* (unwashed) | 1.5 | 0.5 | 679 | 2 | 0 | 0 | 262 | 7 | 264 | 144 |
| *E.coli*(unwashed) | 1.6 | 0.7 | 669 | 55 | 0 | 0 | 209 | 135 | 13 | 257 |
| **Fungal** | | | | | | | | | | |
| Puffball I | 2.4 | 0.7 | 1 | 0 | 0 | 0 | 0 | 0 | 0 | 1 |
| Puffball II | 2 | 0 | 3 | 0 | 0 | 0 | 2 | 0 | 1 | 0 |
| Puffball III | 4.2 | 1.8 | 249 | 98 | 0 | 4 | 46 | 1 | 18 | 82 |
| **Pollen** | | | | | | | | | | |
| Aspen pollen | 5 | 3.2 | 31 | 1 | 0 | 0 | 10 | 2 | 7 | 11 |
| Poplar pollen | 4.4 | 2.9 | 50 | 3 | 0 | 0 | 14 | 1 | 19 | 13 |
| Paper mulberry | 11.4 | 1.4 | 537 | 3 | 0 | 0 | 7 | 0 | 285 | 242 |
| Ryegrass | 3.6 | 2.4 | 15 | 0 | 0 | 0 | 6 | 0 | 3 | 6 |
| **Non-biological** | | | | | | | | | | |
| Fullers' earth | 4.5 | 3.2 | 20 | 9 | 0 | 0 | 4 | 1 | 3 | 3 |
| Phosphate buffered saline | 4.6 | 5.1 | 3 | 2 | 0 | 0 | 0 | 0 | 0 | 1 |
| NaCl | N/A | N/A | 0 | 0 | 0 | 0 | 0 | 0 | 0 | 0 |

**Table C1.** Summary of particle measurements for the 2008 data set using a fluorescent threshold of $3\sigma$.

| Sample | $n > 3\sigma$ | | FL1_280 | FL2_280 | FL2_370 | Size | AF |
|---|---|---|---|---|---|---|---|
| *Bacillus atrophaeus* (unw) | 952 | mean | 94.6 | 63.9 | 65.4 | 1.4 | 7.7 |
| | | S. D. | 47.3 | 36.1 | 44.6 | 0.4 | 3.8 |
| —"— (unw, dil) | 52 | mean | 110.3 | 51.8 | 43 | 1.3 | 8.1 |
| | | S. D. | 84.8 | 79.7 | 76.5 | 0.8 | 6.1 |
| —"— (w) | 1171 | mean | 164.2 | 60.6 | 48.5 | 1.7 | 9.3 |
| | | S. D. | 136.4 | 58.5 | 55.8 | 0.8 | 4.9 |
| —"— (w, dil) | 241 | mean | 140.7 | 50.3 | 46.1 | 1.7 | 9.4 |
| | | S. D. | 97.9 | 57.1 | 58.4 | 0.8 | 5.9 |
| —"— vegetative cells (unw) | 4779 | mean | 239.3 | 221.2 | 192 | 1.1 | 4.7 |
| | | S. D. | 287.5 | 293.3 | 284.9 | 0.4 | 2 |
| —"— —"— (unw, dil) | 1488 | mean | 140.5 | 94.5 | 68.8 | 1 | 6.1 |
| | | S. D. | 71.6 | 70.1 | 60.4 | 0.2 | 3.8 |
| —"— —"— (w) | 1884 | mean | 214.5 | 29.8 | 12.4 | 1.4 | 6.4 |
| | | S. D. | 156.7 | 44.9 | 33.6 | 0.6 | 3.4 |
| —"— —"—(w, dil) | 2064 | mean | 153.8 | 19 | 7.4 | 1.1 | 11 |
| | | S. D. | 43.2 | 19.3 | 17.7 | 0.2 | 5.8 |
| *E coli.* (unw) | 3684 | mean | 222.5 | 240.4 | 247.7 | 1.2 | 4.7 |
| | | S. D. | 301.6 | 351.1 | 375.5 | 0.5 | 2 |
| —"— (unw, dil) | 1448 | mean | 139.3 | 70 | 63.9 | 1 | 5.5 |
| | | S. D. | 99.2 | 56.1 | 59.6 | 0.2 | 2.5 |
| —"— (w) | 2365 | mean | 351.4 | 12.5 | 0.8 | 1.6 | 7.5 |
| | | S. D. | 317.8 | 30.7 | 22.5 | 0.8 | 4.7 |
| —"— (w, dil) | 835 | mean | 202.6 | 10.7 | 4.1 | 1 | 6.6 |
| | | S. D. | 56.3 | 20.2 | 20.8 | 0.2 | 2.8 |
| Bermuda grass | 2681 | mean | 138.1 | 46.5 | 97.9 | 3.6 | 12.6 |
| | | S. D. | 122.7 | 134.7 | 153.9 | 1.2 | 6.4 |
| Johnson grass I | 1209 | mean | 160.5 | 97.4 | 154.9 | 3.5 | 11 |
| | | S. D. | 419.7 | 280.4 | 169.2 | 1 | 6 |
| Johnson grass II | 2673 | mean | 66.7 | 25.7 | 124.4 | 3.3 | 11.6 |
| | | S. D. | 28.9 | 48.3 | 89.4 | 0.8 | 5.7 |
| Birch pollen | 111 | mean | 662 | 433.7 | 250.4 | 4 | 8.3 |
| | | S. D. | 854.8 | 586.9 | 280 | 1.6 | 6.9 |
| Paper mulberry I | 233 | mean | 668.3 | 1228.4 | 1247.9 | 7.3 | 10.9 |
| | | S. D. | 583 | 851.5 | 856.6 | 4.7 | 7.1 |
| Paper mulberry II | 397 | mean | 142.8 | 159.6 | 229.2 | 3.5 | 13.5 |
| | | S. D. | 188.7 | 412.3 | 443.4 | 1.4 | 6.5 |
| Ragweed I | 123 | mean | 384.6 | 219.5 | 173.2 | 3.6 | 10 |
| | | S. D. | 698.7 | 447.9 | 211.1 | 1.3 | 6.5 |
| Ragweed II | 209 | mean | 928.2 | 625.1 | 310.6 | 4.7 | 7.9 |
| | | S. D. | 953.9 | 643.9 | 303.1 | 1.7 | 7.3 |
| Diesel smoke | 11 | mean | 161.1 | 146.5 | 78.4 | 1.2 | 7.8 |
| | | S. D. | 204.3 | 166.2 | 96.3 | 0.3 | 7.5 |
| Grass smoke I | 2542 | mean | 9.8 | 52 | 110.9 | 1.2 | 4.4 |
| | | S. D. | 18.2 | 33.3 | 66.9 | 0.4 | 1.9 |
| Grass smoke II | 815 | mean | 10.9 | 44.2 | 108 | 1.2 | 4.9 |
| | | S. D. | 19.2 | 32.7 | 59.2 | 0.4 | 2.4 |

**Table C2.** Summary of particle measurements for the 2008 data set using a fluorescent threshold of $9\sigma$.

| Sample | $n > 9\sigma$ | statistic | FL1_280 | FL2_280 | FL2_370 | Size | AF |
|---|---|---|---|---|---|---|---|
| *Bacillus atrophaeus*(unw) | 34 | mean | 214.2 | 142.5 | 163.3 | 2.2 | 10.2 |
| | | S. D. | 79.3 | 60.2 | 72.9 | 0.6 | 5.6 |
| —"— (unw, dil) | 4 | mean | 230.8 | 259.2 | 242.5 | 2.7 | 14.4 |
| | | S. D. | 243.8 | 162.1 | 142 | 1.8 | 11.3 |
| —"— (w) | 217 | mean | 358 | 121.6 | 110.1 | 2.6 | 12.4 |
| | | S. D. | 218.1 | 105.4 | 95.5 | 1.1 | 6.1 |
| —"— (w, dil) | 38 | mean | 276 | 128 | 123.7 | 2.6 | 14.8 |
| | | S. D. | 169.6 | 97.3 | 101.8 | 1 | 7.9 |
| —"— vegetative cells (unw) | 1915 | mean | 423.7 | 400.4 | 358 | 1.4 | 4.8 |
| | | S. D. | 384 | 399.3 | 393.3 | 0.6 | 2.4 |
| —"— —"— (unw, dil) | 264 | mean | 244.9 | 166 | 123.1 | 1.2 | 7.5 |
| | | S. D. | 94.5 | 117.3 | 103.9 | 0.3 | 4.6 |
| —"— —"— (w) | 573 | mean | 347.9 | 50.3 | 21.4 | 1.7 | 7.6 |
| | | S. D. | 230.2 | 72 | 53.9 | 0.9 | 3.9 |
| —"— —"—(w, dil) | 194 | mean | 247.7 | 26.8 | 11.1 | 1.3 | 13.8 |
| | | S. D. | 32.6 | 21.4 | 17 | 0.2 | 6.8 |
| *E coli.* (unw) | 1547 | mean | 413.7 | 447.1 | 470.3 | 1.5 | 4.8 |
| | | S. D. | 388.8 | 467.3 | 498.4 | 0.7 | 2.4 |
| —"— (unw, dil) | 371 | mean | 254.1 | 75.4 | 68.8 | 1.1 | 6.6 |
| | | S. D. | 111.1 | 86.8 | 92.2 | 0.2 | 3 |
| —"— (w) | 1461 | mean | 463.6 | 18.1 | 2.8 | 1.8 | 8.5 |
| | | S. D. | 360.6 | 34.3 | 24 | 0.9 | 5.2 |
| —"— (w, dil) | 302 | mean | 260 | 12.8 | 6.4 | 1.1 | 7 |
| | | S. D. | 47 | 22.5 | 24.7 | 0.2 | 3.2 |
| Bermuda grass | 423 | mean | 271.7 | 203.8 | 303.7 | 4 | 13.4 |
| | | S. D. | 262.6 | 285.5 | 303.1 | 1.5 | 7.3 |
| Johnson grass I | 259 | mean | 510.2 | 380.2 | 344.8 | 4.2 | 9 |
| | | S. D. | 814.8 | 513 | 289.3 | 1.3 | 6.1 |
| Johnson grass II | 378 | mean | 77.7 | 82.5 | 267.8 | 3.8 | 12.6 |
| | | S. D. | 41.1 | 96.2 | 161 | 1 | 6 |
| Birch pollen | 56 | mean | 1229.8 | 828.8 | 406.1 | 4.6 | 5.4 |
| | | S. D. | 891.9 | 605.8 | 324.5 | 1.7 | 5.8 |
| Paper mulberry I | 209 | mean | 730.8 | 1363.4 | 1384.5 | 7.9 | 11.2 |
| | | S. D. | 583.7 | 794.4 | 798 | 4.5 | 7.1 |
| Paper mulberry II | 103 | mean | 258.1 | 556 | 690.7 | 4 | 13.4 |
| | | S. D. | 340.3 | 663.8 | 682.1 | 2 | 7 |
| Ragweed I | 34 | mean | 1188 | 750.4 | 393.1 | 4.7 | 6.5 |
| | | S. D. | 933.1 | 577.8 | 299.9 | 1.4 | 6.5 |
| Ragweed II | 117 | mean | 1590.7 | 1089.8 | 484.3 | 5.5 | 5.2 |
| | | S. D. | 791.9 | 499.4 | 307.5 | 1.5 | 6.7 |
| Diesel smoke | 5 | mean | 284.8 | 281.2 | 173.8 | 1.2 | 4.6 |
| | | S. D. | 248.8 | 160.7 | 58.9 | 0.1 | 2.9 |
| Grass smoke I | 231 | mean | 10.9 | 106.4 | 262.7 | 2 | 3.3 |
| | | S. D. | 18.3 | 51 | 127.4 | 0.5 | 2 |
| Grass smoke II | 68 | mean | 8.6 | 102.1 | 260.3 | 2 | 4 |
| | | S. D. | 18.9 | 40.5 | 93 | 0.4 | 3.2 |

**Table C3.** Summary of particle measurements for the 2014 data set using a fluorescent threshold of $3\sigma$.

| Sample | $n > 3\sigma$ | statistic | FL1_280 | FL2_280 | FL2_370 | Size | AF |
|---|---|---|---|---|---|---|---|
| *Bacillus atrophaeus* (unwashed) | 1728 | mean | 104.5 | 45.5 | 26.5 | 1.2 | 8.4 |
| | | S. D. | 118 | 45.9 | 61.2 | 0.4 | 4.3 |
| *Bacillus atrophaeus* (washed) | 1322 | mean | 25.4 | 211.2 | 357 | 1.2 | 5 |
| | | S. D. | 69.5 | 222.7 | 376.5 | 0.5 | 2.1 |
| *E. coli* (unwashed) | 1290 | mean | 104.3 | 174.9 | 317.4 | 1.3 | 6.1 |
| | | S. D. | 187.3 | 207.1 | 395.6 | 0.6 | 2.8 |
| Puffball I | 504 | mean | 288.2 | 218.1 | 169.3 | 3.4 | 12.1 |
| | | S. D. | 524.4 | 289 | 182 | 1.8 | 9.8 |
| Puffball II | 35 | mean | -19.6 | 64.4 | 118.4 | 2.5 | 17.6 |
| | | S. D. | 17.8 | 49.9 | 107.7 | 0.8 | 8.7 |
| Puffball III | 16 | mean | 19.4 | 64.2 | 100.2 | 2.5 | 20.6 |
| | | S. D. | 165.4 | 68.4 | 60.3 | 1.2 | 12.3 |
| Aspen pollen | 74 | mean | 131.3 | 301 | 447.6 | 3.7 | 17.2 |
| | | S. D. | 385.8 | 504.4 | 631.4 | 2.5 | 7.5 |
| Paper mulberry pollen | 541 | mean | 99.9 | 1907.9 | 1924.1 | 11.3 | 11.8 |
| | | S. D. | 77.9 | 311.9 | 260.9 | 1.6 | 5.5 |
| Poplar pollen | 104 | mean | 163.2 | 338.2 | 496.2 | 3.6 | 17 |
| | | S. D. | 488.6 | 525.4 | 643.3 | 2.4 | 9.1 |
| Ryegrass pollen | 21 | mean | 110.7 | 278.7 | 569.3 | 3.3 | 18.4 |
| | | S. D. | 340 | 258.6 | 431 | 2.1 | 8.6 |
| Fullers earth | 61 | mean | 180.2 | 114.3 | 148.2 | 3.7 | 16 |
| | | S. D. | 476.2 | 214.5 | 367.8 | 2.8 | 9.9 |
| NaCl | 3 | mean | 16.7 | 19.7 | 14.7 | 2 | 9.1 |
| | | S. D. | 5.4 | 24.4 | 32.5 | 0.7 | 5.3 |
| Phosphate buffered saline | 35 | mean | 64.2 | 113.9 | 89.1 | 1.4 | 6.2 |
| | | S. D. | 342.1 | 320 | 324.1 | 1.8 | 2.7 |

**Table C4.** Summary of particle measurements for the 2014 data set using a fluorescent threshold of $9\sigma$.

| Sample | $n > 9\sigma$ | statistic | FL1_280 | FL2_280 | FL2_370 | Size | AF |
|---|---|---|---|---|---|---|---|
| *Bacillus atrophaeus* (unwashed) | 684 | mean | 195 | 60.5 | 46.2 | 1.4 | 9.8 |
| | | S. D. | 144.6 | 65.3 | 90.3 | 0.5 | 4.7 |
| *Bacillus atrophaeus* (washed) | 608 | mean | 65.4 | 358.7 | 636.8 | 1.6 | 4.6 |
| | | S. D. | 83.6 | 257.9 | 402.1 | 0.5 | 2 |
| *E. coli* (unwashed) | 632 | mean | 199.9 | 284.1 | 550 | 1.7 | 6.2 |
| | | S. D. | 229.6 | 251.3 | 460 | 0.7 | 3.1 |
| Puffball I | 248 | mean | 599.7 | 380.6 | 252.3 | 4.3 | 8.7 |
| | | S. D. | 606 | 341.3 | 226.1 | 1.8 | 8.2 |
| Puffball II | 3 | mean | -20.7 | 176.7 | 417.3 | 2.4 | 19.7 |
| | | S. D. | 17.4 | 76 | 146.6 | 0.7 | 9 |
| Puffball III | 1 | mean | 654 | 298 | 284 | 2 | 25.6 |
| | | S. D. | 0 | 0 | 0 | 0 | 0 |
| Aspen pollen | 31 | mean | 338 | 643.5 | 952 | 5 | 18.7 |
| | | S. D. | 529.9 | 635.9 | 716.1 | 3.2 | 8.6 |
| Paper mulberry pollen | 537 | mean | 101 | 1921.8 | 1937.7 | 11.4 | 11.8 |
| | | S. D. | 77.2 | 268.5 | 209.1 | 1.4 | 5.5 |
| Poplar pollen | 50 | mean | 355.9 | 644.5 | 938.8 | 4.4 | 16.5 |
| | | S. D. | 651.1 | 626.5 | 694.4 | 2.9 | 9.8 |
| Ryegrass pollen | 15 | mean | 168.1 | 361.5 | 753.3 | 3.6 | 17.8 |
| | | S. D. | 387.7 | 263.5 | 375.9 | 2.4 | 9.5 |
| Fullers earth | 20 | mean | 521.1 | 274.6 | 411.7 | 4.5 | 15.7 |
| | | S. D. | 719.7 | 317.2 | 552.1 | 3.2 | 11.3 |
| Phosphate buffered saline | 3 | mean | 748.7 | 725.7 | 711 | 4.6 | 4.7 |
| | | S. D. | 918.1 | 878.5 | 888.2 | 5.1 | 1.6 |