# Peer review of "Machine learning for improved data analysis of biological aerosol using the WIBS"

_Atmospheric Measurement Techniques, 2018_

## Referee Comment (RC1) · D. Baumgardner (Referee) · 9 Jul 2018

Developing techniques for extracting information from UV-LIF measurements is an important endeavor and the more that can be tested, the better.

The current study follows in the footsteps of an earlier exercise that tested a number of methodologies in the search for optimum approaches that can discriminate PBAP types.

Although the current study appears to have made some additional progress I am of the opinion that there are a number of aspects to the data and the PBAP properties that are missing and that should be included before this study is published.

1) Table I and II are both labeled 2008 2) If Tables I and II are actually 2008 and 2014,

then there needs to be a third table that summarizes the properties that are shown in Figs. 4&5, not only the averages but their variances, as well. These need to be listed for both years for the same test particles because from an examination of the figures, it certainly appears that the properties are quite different for the same biotypes. If this is indeed the case, then it is no surprise that there are different results using the various clustering methods for the two data sets. 3) Why are just FL1,2 and 3 used. In Hernandez et al (not cited here, unfortunately), we found that FL1&2 and FL 1&2&3 are important markers. Leaving them out seems like a loss of useful information. 4) Bioaerosols are by their nature irregular in shape and in their fluorescing. Why isn't the variance also used as a parameter in the clustering?

I think that additional work remains to further separate by general categories within bacteria, fungi and pollen. In our analysis of the lab results we were able to quite clearly separate the three bio types just by fluorescence type and size without any sophisticated machine learning. I would have to assume that this can be improved upon using more sophisticated approaches like the current study.

---

## Referee Comment (RC2) · Anonymous Referee #2 · 12 Jul 2018

Evaluation of the usefulness of analysis techniques for WIBS data, the subject of this paper, is important. However, several problems make the paper unsuitable for publication without significant modifications and clarifications.

1. Reasons for clustering in some cases to 2 or 3 clusters is not clear. Figure 6 with LAB 2008, which illustrates the worst rand index, has only two clusters. Why is a case of HAC with only two clusters shown here? There are quite a few papers in the literature applying HAC to atmospheric aerosol. I can't remember any which clustered down to two. There are 9 samples in the 2008 data set combined into four main categories (bacteria, fungal spores, pollens, and smoke). Wouldn't a reasonable number of clusters be expected to be 9, or somewhere between 4 and 9? I do not see how two clusters makes sense. Also, there are 10 samples in the 2008 data set in four

main categories (bacteria, fungal spores, pollens, earth, and two NaCl samples, with and without phosphate buffer). Wouldn't a reasonable number of clusters be expected to be 10 or close to 10? p.12, line 8-9: "In the worst case scenario two clusters are provided both primarily containing bacteria. Inthis case we can conclude that algorithm has failed to differentiate between any of the biological classes." I don't see how the failure is an intrinsic feature of HAC. The failure, at least in part, seems attributable to the choice to use two clusters. Table 5 shows the bacteria, spores, pollen and non-bio in each of the two clusters for the 2008 data. The discrimination of these clusters is remarkably poor. Why not first cluster to 9 and then show a table such as Table 5 but for the 9 particle types? The same applies to Fig. 4, why force these four sample types into three clusters? The results are confusing enough that I'd recommend showing dendrograms for the clusters of both the 2008 and 2014 data sets, and discussing these dendrograms in relation to the data illustrated in Figs. 4 and 5.

2. In comparing the value of classification/clustering approaches the justification for using differentnumbers of clusters for different methods is not clear. p.15, lines 7-10: "As we did in the previous sections we provide matching matrices of the worst-casescenario and best case scenario when using Gradient Boosting using the current data preparation inTables 8 and 9. In the best case scenario we provide a very good classification with very small errors(AR=0.919)." -In Tables 8 and 9, four clusters (which are the minimum number that makes sense) were used intesting Gradient Boosting, while two or three clusters were used in testing HAC (1 or 2 less than the number of categories compared with) in Tables 4 and 5, and two or three clusters and an additional category for Unclassified were used in testing DBSCAN in Tables 6 and 7. (Table 6 has 2014 data and Table 7 has 2008 data). Because of the use of smaller numbers of clusters than categories for HAC and DBSCAN, but the same number of clusters and categories for Gradient Boosting, I cannot see how these results say anything about the relative value of HAC, DBSCAN and Gradient Boosting. One cannot set the metric based on four categories, do HAC and DBSCAN down to two or three clusters, but generate four categories with Gradient Boosting, and then compare decide

on the better algorithm based on the match results.

3.Why is there such confidence in the assumption that combining into categories is valid and appropriate for deciding between classification schemes? Why is there such a focus on combining all the bacteria into one category, pollens into one category, and fungal spores into one category? Why not differentiate into all the categories measured, test on that, and then combine the results for each to obtain the results for all pollens, etc.? The two smut spore samples (2008) have similar features, but these are different from the puff ball spores (2014), and, as far as I know, very different from the large majority of spore types. Maybe I'm misunderstanding what is done here. The two bacteria used here likely make sense to go into one category. Their FL look similar. I'm assuming the goal is to compare techniques for their capability to help understand atmospheric aerosol. Because of the way the conclusions are stated, this work implies that we can have some confidence that results made on clustering to a category "bacteria" makes sense. However, bacteria that survive in sunlight in the atmosphere tend to be more pigmented than E. coli. How about citing an article such as, Y. Tong and B. Lighthart, Solar Radiation Is Shown to Select for Pigmented Bacteria in the Ambient Outdoor Atmosphere, Photochem Photobiol 1997, pp 103-106, in at least acknowledging that the two bacteria used here are not necessarily representative of bacteria in outdoor air. An explanation of the validity of the bacteria category, while taking into account bacterial pigments and fluorophores such as melanins and carotenoids could be helpful.

4. Is size a useful measurement for classification of all these particle types? Why is size treated as a useful quantity in defining clusters when actual pollens of the species used here have sizes much larger than the sizes used in this study (as indicated in Tables 4 and 5)? It seems that the samples of pollens are of pollen fragments. Is there evidence that the size distributions of pollens and fungal spores used in classification here are similar to those in atmospheric particles? The fungal spores also seem to be fragments. I'll assume the "size" is diameter or some effective diameter for non-

spheres. Then puffball diameter (avg. approx. 2 um in Fig. 5), is less than half the value for puffball spores, as far as I know. I think smut spores are 6 to 9 um, much larger than the 4 um or smaller shown in Fig. 4. The hypothesis that these are fragments of spores seems more likely than that the size calibration is incorrect? Some discussion of the relation to size and ambient sampling for pollen and fungal spores is needed, especially if fragments are the objective or part of the objective. Larger particles of one material should fluoresce more strongly than smaller particles, so I can see the usefulness of size or volume for normalizing the FL. But if the algorithms used here benefit from clustering by size, some papers should be cited on the size distributions of pollen and fungal spore fragments measured in the atmosphere. In any case, the sizes in Figs. 4 and 5 need error bars.

5. Tables showing the same charts as in Figs. 4 and 5, but for the particles which were classified, should be shown for the cases on which the conclusions are based.

6. K-means is mentioned in the abstract, introduction, Section 2.4 and Fig. 1. But are any results shown? I'm not seeing any mention of k-means after section 2.4.

Additional issues

1. There appear to be over 80,000 lab-generated particles in the 2008 dataset and over 20,000 in the 2014 dataset. Why is the fraction-of-particles-classified not part of the criteria for best and worst cases? Is a capability to classify more particles a desired feature in studying atmospheric aerosol? It seems odd that 3/4 to 4/5 of lab-generated test particles are not matched.

2. Error bars or some indication of data variation are needed in Figs. 4 and 5.

3. Why not combine the 2008 and 2014 datasets? Combining would help with the generality of the study and may help make it more realistic and applicable to ambient aerosol. The inorganic samples in 2008 are very different from those in 2014. And there are different pollens (except mulberry) in these two years. The WIBS instruments

used here appear to have different sensitivities for the detectors, different filters (or something else?). But three samples (the two bacteria and mulberry pollen) are in both datasets, and so using the ratios of the measured fluorescences and assuming linearity it should be possible to find multiplication factors for the FL. If it is not possible to combine these datasets, an explanation of why it is not feasible should be presented.

4. FL1, FL2 and FL3 are not defined, and yet they are shown in Figs. 4 and 5. They are important for understanding the data analyzed here. These should be defined, for example in section 2.1 where the "four fluorescence measurements" are described.

5. The justification for omitting FL4, i.e., that some samples saturate, is inadequate.

6. Abstract, line 14-16: "Whilst HAC was able to effectively discriminate between the reference particles, yielding a classification error of only 1.8%, similar results were not obtained when testing on laboratory generated aerosol where the classification error was found to be between 11.5% and 24.2%." This is unclear. Aren't all the particles studied here reference particles, e.g., mulberry pollen, E. coli. Even the smoke from the burning grass is a reference aerosol. I guess reference particle means PSL. How about "reference narrow-size-distribution PSL particles" for clarity.

7. p. 12, line 5: "The adjusted rand score is often quite difficult to interpret . . ." That sounds correct. It is not defined in this paper. Even after looking it up, it is not clear what exactly is being done in this paper, especially when there are n categories and m clusters. A little more should explanation is needed.

8. p. 16, line 10: "It is clear that Hierarchical Agglomerative Clustering certainly has it drawbacks." Almost everything has its drawbacks. But this paper does not demonstrate or clarify drawbacks for HAC, as far as I can understand.

9. How about defining the matching matrix as used here. What is the criterion of the match?

10. The Introduction cites general papers on aerosols and their importance, but the

initial description of machine learning does not. How about a very few relevant citations in the initial ML descriptions.

11. What is fluorescing in the NaCl and NaCl+phosphate samples I and J in Fig. 5? Do pure samples of these fluoresce enough to give the values shown?

―――――――――――――――――

---

## Author Comment (AC1) · 14 Sep 2018

**Response to reviewer comments on amt-2018-126**

Simon Ruske

September 14, 2018

**1 Introduction**

The following document outlines the author's response to the two reviewer comments on the manuscript amt-2018-126. The following colours are used to differentiate between the reviewer comments, the response provided by the authors and the actions taken as a consequence of the comment.

| | |
|---|---|
| Blue | Comments made by the reviewer |
| Black | Response from authors to address comment |
| Red | Action points (AP) that have been taken to address the comment |

**2 Response to Anonymous Referee**

We thank the reviewer for taking the time provide such a thorough review which will aid substantially in developing the manuscript to the standard required for publication.

**2.1 Main comments**

**2.1.1 Why have you clustered down to two or three clusters?**

*"1. Reasons for clustering in some cases to 2 or 3 clusters is not clear. Figure 6 with LAB 2008 which illustrates the worst rand index, has only two clusters. Why is a case of HAC with only two clusters shown here? There are quite a few papers in the literature applying HAC to atmospheric aerosol. I can't remember any which clustered down to two. There are 9 samples in the 2008 data set combined into four main categories (bacteria, fungal spores, pollen, and smoke). Wouldn't a reasonable number of clusters be expected to be 9 or somewhere between 4 and 9? I do not see how two clusters makes sense. Also, there are 10 samples in the 2008 data set in four main categories (bacteria, fungal spores, pollen, earth and two NaCl samples, with and without phosphate buffer). Wouldn't a reasonable number of clusters be expected to be 10 or close to 10? p.12, line 8-9 "In the worst case scenario two clusters are provided both primarily containing bacteria. In this case we can conclude the algorithm has failed to differentiate between any of the biological classes." I don't see how the failure is an intrinsic feature of HAC. The failure, at least in part seems attributable to the choice to use two clusters. Table 5 show the bacteria, spores, pollen and non-bio in each of the two clusters for the 2008 data. The discrimination of these clusters is remarkably poor. Why not first cluster to 9 and then show a table such as Table 5 but for the 9 particle types? The same applies to Fig. 4 why force these four sample types into three clusters? The results are confusing enough that I'd recommend showing dendrograms for the clusters of both 2008 and 2014 data sets, and discussing these dendrograms in relation to the data illustrated in Figs. 4 and 5."*

We believe this confusion has arisen from the author's inadequate description of Figure 6, which we detail more thoroughly below. Since the response to this comment is substantial we have provided bullet points summarising the response, followed by a more detail explanation.

- A two cluster solution is presented since this is the solution for which the maximum of the CH index was attained.

- Clusterings containing between 1 and 10 clusters have been produced, but this was not made clear in our description of Figure 6.

- Figure 6 shows the maximum value of the adjusted rand score for clusterings containing between 1 and 10 clusters and the adjusted rand score for the clustering which produced the highest value of the Calinski-Harabasz index.

- A dendrogram alongside a heat map of the matching matrix for the clustering containing 10 clusters has been produced. From the plot we see material from two different broad categories e.g. fungal and pollen has been grouped together prior to the final stages of the hierarchy.

Some of the potential sources of error when using HAC are: errors due to the hierarchical agglomerative clustering routine, errors due to the clustering index used to determine the number of clusters and errors due to the data preparation used prior to the algorithm being applied. Figure 6 has been created in an attempt to differentiate between these errors.

Prior to HAC being applied, we label each particle depending on the broad category (1 for bacteria, 2 for fungal, 3 for pollen and 4 for non-biological). Hierarchical agglomerative clustering is then ran on every possible combination of the considerations provided in Table 3, producing a total of 96 hierarchies.

Subsequently, the clusterings containing between 1 and 10 clusters are extracted from each hierarchy and two statistics are calculated. First, the adjusted rand score is calculated as a measure of how similar each clustering is to the known labels. This measure is intended to provide an indication of performance and would be unavailable during an ambient campaign as cluster membership would not be known. We also calculate the Calinski-Harabasz index (CH index) for each of the clusterings containing between 1 and 10 clusters. This is a statistic usually calculated to determine the number of clusters for data collected in an ambient campaign. In Figure 6, we then present two values of the adjusted rand score. First, the maximum value of the adjusted rand score across the first 10 clusterings (presented in the dark bars). Second, the value of the adjusted rand score in the case of the clustering for which a maximum value of the CH index was obtained (in the light bars). This is intended to demonstrate errors that arise due to the CH index. For example, on the laboratory data collected in 2014 the CH index attains a maximum for the 3 cluster solution (shown in the light bar) and the maximum adjusted rand score was attained also for the 3 cluster solution (shown in the dark bar). This is an example where the CH index has worked at intended attaining a maximum for the clustering which is most similar to the known labels.

Scores are presented when using the data preparation strategy suggested in Crawford et al. (2015) modified to use a fluorescent threshold of either 3 (in blue) or 9 (in orange) standard deviations above the average forced trigger measurement as first suggested in Savage et al. (2017). The green bars show the best result across all 96 strategies tested.

In some of the cases presented in Figure 6, the adjusted rand score and calinski-harabasz attained a maximum for different clusterings. For example, in 2008 when using a fluorescent threshold of 3 standard deviations above the forced trigger measurement, we see that a maximum of the CH index was attained for the 4 cluster solution whereas the most similar clustering to the known labels was for the 5 cluster solution. In this case, the 4 cluster solution was nearly as similar to the known labels as the 5 cluster solution, so concluding that there are 4 clusters instead of 5 is reasonable since the 5 cluster solution was only marginally better than the 4 cluster solution.

However, in 2008 when using a fluorescent threshold of 9 standard deviations above the average forced trigger measurement, poor performance is observed where the CH index attains a maximum for the 2 cluster solution when the most similar clustering to the known labels was the 5 cluster solution. For this specific case, part of the poor performance is due to the CH index attaining a maximum at 2 clusters. But we also see that largest adjusted rand score for clusterings with between 1 and 10 clusters (presented in the dark bar) is still quite low. So the conclusion is that better performance could be attained if the index used concluded there were 5 clusters, but also better performance could be obtained should a different data preparation

strategy be used. The maximum of the adjusted rand score across all data preparation approaches tested (shown in the green bars) for 2008 however is still approximately 0.6 whereas for the same data Gradient Boosting attained an adjusted rand score of nearly 0.9.

To conclude, part of the poor performance is due to the index used to determine the number of clusters, part is in the choice of data preparation used, though we failed to provide a recommendation for a data preparation technique which performed consistently well across all the data sets tested, and part is due to selecting to use the HAC algorithm rather than one of the other the algorithms tested.

We have investigated the tendency of the CH index to conclude that there are 2 clusters in further detail and found that it is possibly due to a larger proportion of the data set consisting of bacterial samples. To demonstrate this point further we have simulated data from three normally distributed clusters centred around (0, 0), (5, 5) and (10, 10) and varied the proportion of the material placed in the cluster with the largest number of particles. As the proportion of the data which is sampled from the largest cluster increases, there is a point where the likelihood of the CH index to make the correct conclusion sharply drops. The proportion of data placed in the largest cluster before this sharp drop occurs is dependent upon the variability of the clusters. If we set the standard deviation of each of the clusters to $\sigma = 3$, it would only require approximately 70% of the data to be from one cluster, before this sharp drop in the accuracy of the CH index occurs. In an ambient environment where we would expect concentrations of bacteria to be an order of magnitude greater than the concentrations of fungal spores, this tendency of the CH index to conclude two clusters could be a significant disadvantage and investigating alternative indices for determining the number of clusters may be required in future studies.

We agree that dendrograms would aid in interpreting these results and the reviewers suggestion for providing Table 5 but for the 10 cluster solution has also been added to the same figure.

**AP1** Produced dendrogram plots alongside a heat map of the matching matrix comparing the 10 cluster solutions to the known labels.

**AP2** Rewritten the HAC results section to make clearer what analysis has been conducted and more adequately describe Figure 6.

**AP3** Produced an additional section to indicate why the CH index has a tendency to conclude that there are 2 clusters.

**AP4** Split the HAC section into subsections highlighting potential considerations for data preparation, the CH index, and potential issues in the hierarchy that could not be rectified by the selection of a different index to determine the number of clusters.

**2.1.2 Why is it valid to cluster down to 2 or 3 clusters for some algorithms but not all?**

*"2. In comparing the value of classification/clustering approaches the justification for using different number of clusters for different methods is not clear. p.15, lines 7-10: "As we did in the previous sections we provide matching matrices of the worst case scenario and best case scenario when using Gradient Boosting using the current data preparation in Tables 8 and 9. In the best case scenario we provide a very good classification with very small errors (AR=0.919)." In tables 8 and 9, four clusters (which are the minimum number that makes sense) were used in testing Gradient Boosting, while two or three clusters were used in testing HAC (1 or 2 less than the number of categories compared with) in Tables 4 and 5, and two or three clusters and an additional category for Unclassified were used in testing DBSCAN in Tables 6 and 7. (Table 6 has 2014 and Table 7 has 2008 data). Because of the use of smaller number of clusters than categories for HAC and DBSCAN, but the same number of clusters and categories for Gradient Boosting, I cannot see how these results say anything about the relative value of HAC, DBSCAN and Gradient Boosting. One cannot set the metric based on four categories, do HAC and DBSCAN down to two or three clusters, but generate four categories with Gradient Boosting, and then compare decide on the better algorithm based on the matched results."*

We thank the reviewer for bringing the lack of clarity regarding how the number of clusters was set for each algorithm to our attention. Dependent on which algorithm is used the number of clusters is set in a different way.

In the case of HAC, the number of clusters has been determined by finding a maximum value of the CH index for the clusterings from 1-10. Since this is what would be used when analysing ambient data it is important to present the 2 and 3 cluster solutions since these solutions would be obtained if we analysed the laboratory data in exactly the same fashion as we would analyse ambient data.

In the case of DBSCAN, we have no direct control over the number of clusters as this is determined indirectly by the choice of epsilon and the number of points required to form a neighbourhood. We have performed a grid search for these parameters, testing a variety of combinations, and there are a number of selections for which a result with more than 3 clusters were produced. However, we attempted to select epsilon and the minimum number of points by inspection of Figure 7 on the basis of which parameters resulted in a consistent performance across the data sets tested and these selections resulted in 2-3 clusters.

Irregardless of the algorithm used, we have tested a fluorescent threshold of both 3 and 9 standard deviations above the average forced trigger measurement. The purpose of such threshold is to remove the instrument noise. Since the vast majority of the non-biological samples in 2014 will fail to exceed these thresholds and hence have been removed, we believe a 3 cluster solution would be reasonable for the 2014 data.

It is important to note in the case of HAC we have tested 96 combinations of data preparations, produced hierarchies for each, calculated the adjusted rand score for the clusterings between 1 and 10 clusters and at no point did we attain an adjusted rand score of greater than 0.75 for the laboratory generated aerosol collected in 2008 and 2014 (this can be seen in Figure 6). However, when using DBSCAN, with the exception of the 3 sigma threshold on the 2008 data, we were able to obtain an adjusted rand score of greater than 0.8 after removing between 25 and 35% of the data (as indicated in Figure 7). In the case of the Gradient Boosting algorithm we consistently attain an adjusted rand score of greater than 0.8. We believe that this should give a strong indication of the potential of considering alternatives to HAC for this particular application.

**AP5** Produce a table that highlights the potential advantages and disadvantages highlighted within the current study and how the numbers of clusters are determined in each case.

**2.1.3 Why combine the data into four categories?**

*"3. Why is there such confidence in the assumption that combining into categories is valid and appropriate for deciding between classification schemes? Why is there such a focus on combining all the bacteria into one category, pollens into one category, and fungal spores into one category? Why not differentiate into all the categories measured, test on that and then combine the results for each to obtain the results for all the pollens etc.? The two smut spore samples (2008) have similar features, but these are different from the puff ball spores (2014), and as far as I know, very different from the large majority of spore types. Maybe I'm misunderstanding what is done here. The two bacteria used here likely make sense to go into one category. Their FL look similar. I'm assuming the goal is to compare techniques for their capability to help understand atmospheric aerosol. Because of the way the conclusions are stated, this work implies that we can have some confidence that results made on clustering to a category "bacteria" makes sense. However, bacteria that survive in sunlight in the atmosphere tend to be more pigmented than E. coli. How about citing an article such as, Y. Tong and B. Lighthart, Solar Radiation is Shown to Select for Pigemented bacteria in the Ambient Outdoor Atmosphere, Photchem Photobiol 1997, pp 103-106, in at least acknowledging the two bacteria used here are not necessarily representative of bacteria in outdoor air. An explanation of the validity of the bacteria category, while taking into account bacterial pigments and fluorophores such as melanins and carotenoids could be helpful."*

There are varying levels of complexity when attempting to discriminate between biological aerosol. In order of difficulty: 1) to be able to discriminate between biological and non-biological material 2) to be able to discriminate between the broad pollen, bacteria and fungal categories and 3) to be able to discriminate

between different species of pollen, bacteria and fungal spores. If an algorithm was capable of discriminating between the different samples, it seems logical that the algorithm would be capable of discriminating between the broad biological classes. Similarly if an algorithm could not discriminate between the broad biological classes we would believe that the algorithm would not be able to discriminate between the individual samples.

After producing the dendrograms suggested and matching matrices of the ten cluster solutions against the samples as suggested by the reviewer in Section 2.1.1, it has become more apparent that fungal material is being grouped with pollen material (2008) or with bacterial material (2014) prior to the final stages of the algorithm. So we do not believe HAC is not segregating by sample either. Instead, fungal material is being grouped with other classes prior to the final stages of HAC.

Given that there are still significant errors in classifying between fungal and pollen samples irregardless of algorithm used, we would suggest that attempting to discriminate between individual samples may be more successful using more recently developed instruments such as the MBS and the WIBS NEO or in future analyses where a wider variety of samples are collected.

We have read Tong and Lighthart (1997) which does indicate that pigmented bacteria is more prevalent in the presence of solar radiation, increasing to $50 - 60\%$ at noontime compared to approximately $33\%$ at midnight. That being said, unless we are misunderstanding the article, these findings seem to indicate that non-pigmented bacteria do constitute between $40\%$ and $67\%$ of outdoor air dependent on time of the time of day. So, if pigmentation did significantly change the fluorescence response from the instrumentation it would seem that collection of both pigmented and non-pigmented bacteria would be required to characterise outdoor air.

A comment regarding the description of fluorophores within the current paper has also been made during the technical review stage, where it has been noted that citing Pöhlker et al. (2013) may be helpful. We also found an earlier article by the same author (Pöhlker et al., 2012), which we believe may also be of value in investigating the potential influence of pigments such as melanin and carotenoids.

In Table 1 in Pöhlker et al. (2012), a wide range of atmospherically relevant biological fluorophores are summarised, including a number of pigments. The excitation wavelengths reported for melanin and carotenoids, $469 - 471$nm and $400 - 500$nm respectively, are different to the excitation used by the WIBS which is 280nm and 370nm. Melanin is also noted to have relatively low fluorescence intensity and is estimated to have low relevance for fluorescent biological aerosol particle (FBAP) detection. Despite Pöhlker et al. (2012) suggesting that the role of carotenoids in FBAP detection is high, there are articles on the fluorescence of carotenoids that report very low fluorescence (Gillbro and Cogdell, 1989). It therefore does appear that carotenoids and melanins probably do not significantly influence the fluorescence response from the WIBS.

To further investigate pigmentation we also read an article specific to Bacillii (Khaneja et al., 2010). Colonies of *Bacillus atrophaeus* (one of the samples presented in the current study) were presented that appeared yellow-orange and others which appeared grey, although carotenoid production in the yellow-orange samples was low. *Bacillus subtilis*, which has been presented in Hernandez et al. (2016), is noted to carry a melanin-like compound to protect against solar radiation but shows very similar fluorescence response to the *E. coli* sample also presented in Hernandez et al. (2016). Given that a significant proportion of the bacterial content in the UK is believed to be Bacillii (Harrison et al., 2005), the inclusion of *Bacillus atrophaeus* as one of our samples seems sensible.

Whether additional bacteria with more varied pigmentation will be required to characterise the outdoor environment is an issue that could be investigated more thoroughly in further research through the collection of a wider range of bacteria and, if possible, by measuring the instrument response to pigments. Whilst the impact of pigmentation in bacteria cannot be fully addressed at this point in time, we recognise that our current description of fluorophores is lacking, and have attempted to make improvements in response to this comment.

**AP6** Updated the description of the fluorophores in the introduction to include additional references.

**2.1.4   Is size a useful measurement?**

*"4. Is size a useful measurement for classification of all these particle types? Why is size treated as a useful quantity in defining clusters when actual pollens of the species used here have sizes much larger than the sizes used in the study (as indicated in Tables 4 and 5)? It seems that the samples of pollens are of pollen fragments. Is there evidence that the size distributions of the pollens and fungal spores used in classification here are similar to those in atmospheric particles? The fungal spores also seem to be fragments. I'll assume the "size" is diameter or some effective diameter for non-spheres. Then puffball diameter (avg. approx. 2 um in Fig. 5), is less than half the value for puffball spores, as far as I know. I think smut spores are 6 to 9 um, much larger than the 4 um or smaller shown in Fig. 4. The hypothesis that these are fragments of spores seems more likely than the size calibration is incorrect? Some discussion of the relation to size and ambient sampling for pollen and fungal spores is needed, especially if fragments are the objective or part of the objective. Larger particles of one material should fluoresce more strongly than smaller particles, so I can see the usefulness of size or volume for normalising the FL. But if the algorithms used here benefit from clustering by size, some papers should be cited on the size distributions of pollen and fungal spore fragments measured in the atmosphere. In any case, the sizes in Fig. 4 and 5 need error bars."*

We feel that size is definitely a useful measurement for all of these particle types. To better understand the relationship between size and fluorescence for each of the samples collected we have produced fluorescence versus size on scatter plots which we intend to include in the supplementary material. As we did in Section 2.1.1, we provide summary bullet points followed by a more thorough response.

- The WIBS measurement of size is different to other size measurements, but we feel that it is useful in discriminating between the particles collected.

- Whether the samples are fragments could be more thoroughly investigated in future by using microscopy in future experiments, but we agree with the hypothesis that a large quantity of the pollen samples collected are likely fragments.

- We have included additional references to compare the size ranges in the current study to other studies using the WIBS as well as other studies which use microscopy.

- As newer versions of the instrument are developed, measurements for larger particles, including more intact pollen will become possible.

The WIBS size measurement is an optical scatter calculation, calibrated with unit density polystyrene latex (PSL) spheres. If the particles are a different density (dry pollen) or refractive index (soots), or irregular in shape (dusts, clumps etc) then the resultant size measurement will likely be different from alternative measurements such as those from viewing the particles under a microscope. Whether or not the particles are fragments or not could be more thoroughly investigated through future research, i.e. by collecting filter samples of the particles after they are placed into the chamber.

We have now included several additional citations on the size ranges expected for the samples collected. Fortunately, samples of the same or similar species have previously been collected using the WIBS instrumentation across a number of studies (Healy et al., 2012; Hernandez et al., 2016; Savage et al., 2017). In Healy et al. (2012) a size range of $\sim 3 - 30 \mu$m (low gain) is used when collecting pollen samples, whereas for the fungal samples they collected they used high-gain mode ($\sim 0.5 - 12 \mu$m). The "low gain" mode, is available in the WIBS version 4 but the WIBS version 3 which is used to collect the data presented is limited to an approximate size range of between 0.5 and 12 microns according to Healy et al. (2012), although we did measure particles as large as 14 microns. In Hernandez et al. (2016), low-gain was used for the fungal and the pollen samples whereas the high-gain mode is used for the bacterial samples. In Savage et al. (2017), microscopy is used to support the hypothesis of a mixture of intact pollen and fragmented pollen being present in the samples collected. The size ranges for the pollen samples, with the exception of the mulberry sample, presented in the current study are similar to those presented in Hernandez et al. (2016) and would be consistent with the hypothesis that the sample is comprised mostly, if not entirely, of fragmented pollen. The mulberry sample has been also analysed in Healy et al. (2012) where an average size of $13.6 \pm 6.2 \mu$m is presented, very similar to the value of $13.8 \mu$m which has been presented in other studies using microscopy

(Kang et al., 2007). The sizes of the paper mulberry samples presented here are $7.18 \pm 4.74$ and $3.40 \pm 1.42$ for two of the sample files from 2008 and then $11.27 \pm 1.74$ for the one sample file collected in 2014. For the first sample of Paper Mulberry collected in 2008, it would seem that we have a mixture of intact and fragmented pollen, whereas in the second sample could be entirely comprising of pollen fragments. In 2014, we may be viewing a sample consisting of primarily intact pollen, albeit only the smaller tail of the size distribution presented in Healy et al. (2012).

Inspecting the scatter plots of size versus fluorescence (which will be placed in the supplementary material for the re-submission), we can see in the case of one of the puffball samples there a number of particles just above the fluorescent threshold for a wide size range. However, there is a clear cluster of particles well above the fluorescent threshold for one particular file with a much narrower size range around 6 microns. A similar pattern is apparent in one of the Johnson-grass smut samples. The size range of the Bermuda grass sample is similar to the sizes presented in Healy et al. (2012) whereas the size range of the Johnson grass is substantially smaller. In both the current study and Healy et al. (2012) the size ranges are smaller than the dimensions quoted in a microscopy study on fungal smuts (Crotzer and Levetin, 1996). So it does seem possible that these samples will contain at least some fragments. The puff ball spores have been studied previously using a fluorescence particle counter where the size range was stated to be between 2-4$\mu$m for the particles that they believed to be puffball spores. Two of the puffball sample files produced in the current study had average particle size of $2.50\pm0.85$ and $2.45\pm1.16$, but only 35 and 16 measurements from these files exceeded a fluorescent threshold of three standard deviations above the average forced trigger measurement. Whereas the other puffball sample collected had a size range of $3.39 \pm 1.76\mu$m with 506 particles exceeding the $3\sigma$ threshold.

We recognise that sizes of the pollen collected may be different to those in atmospheric particles. However, the data collected should be representative, at least to some extent, of what would be collected using the instrument during an ambient campaign. Conclusions from this data should provide an enhancement of conclusions stated in Crawford et al. (2015) where PSLs alone have been used to inform our analysis approach.

Since there is a large amount of historic data, and measurements are still being collected using the same instrument, we believe the findings presented will be valuable at this point in time. Nonetheless, as newer instruments are developed for example the WIBS NEO from Droplet Measurement Technologies, particles will be collected over a larger size range which will likely include a larger quantity of intact pollen. We expect such measurements will be more representative of an atmospheric environment and the current study should also be somewhat helpful as starting point for future research using instrumentation that has been more recently developed.

**AP7** Added a table in the appendix section comparing the average size of the particles presented with other studies.

**AP8** Add error bars to size in Figures 4 and 5.

**2.1.5 Add fluorescence tables for results**

*"5. Tables showing the same charts as in Figs. 4 and 5, but for the particles which were classified, should be shown for the cases on which the conclusions are based."*

We thank the reviewer for this suggestion which will improve the paper. We have added tables of the average measurements of the particles classified to the appendix.

**AP9** Added the tables suggested.

**2.1.6 What happened to k-means?**

*"6. K-means is mentioned in the abstract, introduction, Section 2.4 and Fig. 1. But are any results shown? I'm not seeing any mention of k-means after section 2.4."*

The results for K-means were generally very poor. We have added a sentence to the main text to indicate that this was the case and add further details to supplementary material.

**AP10** Added sentence to describe the poor performance of k-means to the text and add details to the supplementary material.

**2.2 Other Issues**

**2.2.1 Why is fraction of particles not used as a criteria for good performance**

*"1. There appear to be over 80, 000 lab-generated particles in the 2008 dataset and over 20,000 in the 2014 dataset. Why is the fraction-of-particles-classified not part of the criteria for best and worst cases? Is a capability to classify more particles a desired feature in studying atmospheric aerosol? It seems odd that 3/4 to 4/5 of lab-generated test particles are not matched."*

There seems to be some level of instrument noise present in the samples that should be removed by the threshold imposed. Such measurements are definitely worth removing. The only algorithm which removes any additional particles not already removed by the fluorescent and size threshold is DBSCAN. But whether this is an advantage or disadvantage is rather subjective. None of the algorithms worked perfectly on the data tested. If it is the case that a particle cannot be correctly classified, whether it is better to classify the particle incorrectly or not classify it at all is debatable.

In any case, in similar studies, (e.g. Hernandez et al., 2016), a similarly large number of particles from a sample do not fluoresce in any of the channels and are removed. For example 19786 particles are collected for the *Bacillus subtilis* sample with only 100 particles being fluorescent in at least one channel.

**2.2.2 Error bars or some indication of data variation are needed in Figs. 4 and 5**

This issue has also been raised by Darrell Baumgardner and is addressed in our response to this review.

**2.2.3 Why not combine the 2008 and 2014 datasets?**

*Why not combine the 2008 and 2014 datasets? Combining would help with the generality of the study and may help make it more realistic and applicable to ambient aerosol. The inorganic samples in 2008 are very different from those in 2014. And there are different pollens (except mulberry) in these two years. The WIBS instruments used here appear to have different sensitivities for the detectors, different filters (or something else?). But three sample (the two bacteria and mulberry pollen) are in both datasets, and so using the ratios of the measured fluorescences and assuming linearity it should be possible to find multiplication factors for the FL. If it is not possible to combine these datasets, an explanation of why it is not feasible should be presented.*

We would agree that combining the data sets would perhaps be valuable. But on closer inspection of the data we see that the paper mulberry samples collected had different size ranges across the two years. It therefore would be quite difficult to combine the data sets as suggested. There is also the possibility that average forced trigger fluorescence measurements could be subtracted from each of the sample measurements in an attempt to combine the files. However, investigating whether data could be combined in such a fashion, would be more appropriate once further data is collected alongside measurements using other techniques such as microscopy, whereby we could be more certain of what particles are being measured by the instrument.

Furthermore, one of the findings from the study is that the conclusions that one makes as to how to prepare the data for HAC is dependent upon what data is used to make these conclusions. So keeping the data sets separate, may be beneficial in highlighting the importance of repeating experiments.

**AP1** Update text to describe why the data sets have not been combined at this point

**2.2.4    Define FL1, FL2 and FL3**

*"4. FL1, FL2 and FL3 are not defined, and yet they are shown in Figs. 4 and 5. They are important for understanding the data analysed here. These should be defined, for example in section 2.1 where the "four fluorescent measurements" are described."*

The FL1, FL2 and FL3 notation has been used previously in other studies (e.g Healy et al., 2012), but may cause confusion between the current study and the notation used in Kaye et al. (2005) so has been replaced with $FL1\_280$, $FL2\_280$ and $FL2\_370$ respectively.

**AP2** Updated text and figures to include alternative notation and define these in the data section.

**2.2.5    Justify fully why you have omitted FL4.**

*"The justification for omitting FL4, i.e., that some particles saturate, is inadequate"*

Consulting Kaye et al. (2005), the following description should be more appropriate.

"The particle is irradiated with UV light at 280nm and 370nm from the firing of two xenon sources. Fluorescence emission is collected via two collection channels in the ranges $310 - 400$nm and $420 - 600$nm. The 370nm xenon radiation lies within the first detection range and hence elastically scattered light from the particle, sufficient to saturate the detection amplifier, is received. This saturated signal is therefore discarded. "

**AP3** Added this description to the text.

**2.2.6    "Reference Particles"**

*"6. Abstract, line 14-16: "Whilst HAC was able to effectively discriminate between the reference particles, yielding a classification error of only 1.8%, similar results were not obtained when testing on laboratory generated aerosol where the classification error was found to be between 11.5% and 24.2%." This is unclear. Aren't all the particles studied here reference particles, e.g., mulberry pollen, E. coli. Even the smoke from the burning grass is a reference aerosol. I guess reference particle means PSL. How about "reference narrow-size distribution PSL particles" for clarity"*

We thank the reviewer for this suggestion and agree that the wording proposed is clearer so have used this instead.

**AP4** Add the suggestion

**2.2.7    Describe Adjusted Rand Score**

*"p. 12 line 5: "The adjusted rand score is often quite difficult to interpret ..." That sounds correct. It is not defined in this paper. Even after looking it up, it is not clear what exactly is being done in this paper, especially when there are n categories and m clusters. A little more explanation is needed."*

See response to 10.

**2.2.8    Clarify drawbacks**

*"8. p. 16, line 10: "It is clear that Hierarchical Agglomerative Clustering certainly has it drawbacks." Almost everything has its drawbacks. But this paper does not demonstrate or clarify drawbacks for HAC, as far as I can understand."*

We have added a table to the text to make this clearer to the reader, including potential advantages and disadvantages of the other algorithms which may not be apparent in the current submission.

**2.2.9   Define the matching matrix**

See response to 10.

**2.2.10   Add some references for ML**

*"10. The introduction cites general papers on aerosols and their importance, but the initial description of machine learning does not. How about a very few relevant citations in the initial ML descriptions."*
We thank the reviewer for this suggestion. We agree that inclusion of these references alongside a description of the adjusted rand score and the matching matrices would be useful.

**AP5** Add references suggested and additional information to the methods section.

**2.2.11   What is fluorescent in I & J?**

*"What is fluorescing in the NaCl and NaCl+phosphate samples I and J in Fig. 5? Do pure samples of these fluoresce enough to give the values shown?"*
The plot currently presented is of the fluorescence measurements after a threshold has been applied. In the case of the NaCl and NaCl+phosphate samples these measurements would be for 3 and 61 particles respectively. These particles are likely to be low level contamination. On reflection we realise that presenting this information is of little value to the readership so we have removed it. In addition we realise, especially in the case of the gradient boosting algorithm it would not be reasonable to train on this class and as such this has been removed from the analysis.

For the data collected in 2008, the diesel soot and grass smoke samples could be expected to fluoresce and it may be beneficial to be able to discriminate between the fluorescent particles within this sample and the remainder of the data so these particles remain in the analysis.

**AP6** Remove salt samples from plot.

**AP7** Rerun analysis using gradient boosting without the salt samples.

**3   Response to Darrell Baumgardner**

We would like to thank Darrell for taking the time to review our manuscript. Many of the comments provided highlighted some issues that we had not previously considered and will aid in improving the paper to publication standard. We would like to apologise for initially not including a citation for the Hernandez study in the submission. This choice was made in an attempt to keep the message of the paper succinct, but upon reflection we realise that its inclusion would aid improving the paper, not only in contextualising the findings but to compare and contrast results with previous studies.

**3.1   Incorrect labelling of Tables**

*1) Table I and II are both labelled 2008*
We thank you for bringing this incorrect labelling to our attention. This had been rectified during the technical review stage, but we believe you may be reading the previous version of the manuscript prior to the technical review. Nonetheless, it should be 2008 for Table I and 2014 for Table II and we will ensure that this labelling is correct in the revised submission.

**3.2 Need to summarise variation in the data**

*"2) If Tables I and II are actually 2008 and 2014, then there needs to be a third table that summarizes the properties that are shown in Figs. 4&5, not only the averages but their variances as well. These need to be listed for both years for the same test particles because from an examination of the figures, it certainly appears that the properties are quite different for the same biotypes. If this is indeed the case, then it is no surprise that there are different results using the various different clustering methods for the two data sets."*

The variation of the data does need to be explored in more detail. There are differences in the fluorescent properties that can be explained by differences in sample preparation. In particular, for the bacteria there are unwashed and washed samples, diluted and undiluted samples and some samples of vegetative cells. As a result we believe that Figures 4 & 5 do need re-plotting to segregate further by sample. We have also produced scatter plots of fluorescence against size in each of the three channels. We are considering providing some of these plots in the main text with a link to similar plots for the remaining samples. In addition, we have included the table you have suggested alongside ABC counts, similar to the table presented in the appendix of Hernandez et al. (2016), which should aid in comparing the studies.

You are right that a potential explanation of the algorithmic performance for the two data sets could be that there are differences in the fluorescence properties in each case. In addition, the different thresholds used would result in a difference proportion of the different samples being present in the data set tested which could also affect the performance. This consideration has been added to the main text.

However, it should be at least a slight concern that the performance of the unsupervised techniques seems to be dependent on what data they were tested on, as we would hope that HAC would be adaptable to a variety of different situations. Gradient boosting, the supervised technique tested, did provide a smaller classification error across all of the tests and did seem to perform well consistently across the variety of tests conducted, so long as a fluorescent threshold of either 3 or 9 standard deviations was applied.

**AP1** Produce the table requested.

**AP2** Update Figures 4 and 5.

**3.3 Why not use the biomarkers suggested in Hernandez?**

*"3) Why are just FL1, 2 and 3 used. In the Hernandez study (not cited here, unfortunately), we found that FL1 & 2 and FL 1 & 2 & 3 are important markers. Leaving them out seems like a loss of useful information.*

We are using the raw fluorescent measurements when conducting our analysis, so information would not be lost in this way. When a single decision tree is fit to the data all possible splits are considered, including the splits using the thresholds defined in the Hernandez study, and as such the performance of a single decision tree cannot be worse than the approach suggested in the Hernandez study. So when using decision and ensembles of decision trees information will not be lost as suggested.

In the case of the unsupervised algorithms it is indeed an interesting idea to see whether better performance could be attained by clustering the biomarkers indicated in the Hernandez study rather than the raw data. However at this point we would be informing our analysis using laboratory data, so arguably the analysis would cease to be unsupervised.

**3.4 Why not use the variance?**

*"4) Bioaerosols are by their nature irregular in shape and in their fluorescing. Why isn't the variance also used as a parameter in the clustering"*

We believe you are referring to either to the variance of each broad class or the variance of the samples. The variance could be used in the clustering of laboratory data, but during an ambient campaign we would not know the classification of each sample so the variance of each group or specific particle type could not be explicitly calculated.

**3.5  More work?**

*"I think additional work remains to further separate by general categories within bacteria, fungi and pollen. In our analysis of the lab results we were able to quite clearly separate the bio types just by fluorescence and size without any sophisticated machine learning. I would assume that this can be improved upon using more sophisticated approaches like the current study."*

To construct our response we have use two papers (Hernandez et al., 2016; Calvo et al., 2018). For the benefit of other readers we briefly describe these papers. First, a particle may be described as A, B or C if they exceed the fluorescent threshold in the first, second or third fluorescent channels respectively. These ABC labels can then be combined to make groups of A, B, AB, C, AB, AC and ABC. For example, "AB" would describe particles that exceeded the threshold in the first and second channel.

The grouping of the data in such a fashion, referred to as "ABC analysis", was first introduced in Hernandez et al. (2016) and has since been applied to ambient data in Calvo et al. (2018). In Calvo et al. (2018) more detail is provided on how these ABC counts could be combined with the equivalent optical diameter (EOD) to provide a classification of WIBS data e.g. "Type I: Having the characteristics of the library bacteria (category A or AB, EOD $< 1.5\mu$m)."

From Figure 3 in Hernandez et al. (2016) the average measurements of each of the samples can clearly be divided. However, such a plot does not include the variation of the data. If one further investigates Table A1 in Hernandez et al. (2016), there are some particles that are fungi for example that have fluorescence type ABC that may be incorrectly classified using the classification scheme suggested in Calvo et al. (2018), if they are larger than $2\mu$m,. In addition, we believe if the size of the bacteria is log normally distributed with the mean and standard deviation presented in Hernandez et al. (2016) that some of the particles will exceed 1.5 microns which is the threshold set in Calvo et al. (2018) for the bacteria.

Without having access to the full data set from this study it is difficult to determine precisely what proportion of the data will be incorrectly classified using such an approach, to directly compare with the techniques tested in this study. However, we do see the value in the application of the approach suggested in Hernandez et al. (2016), and results using ABC analysis for the data presented has been conducted and added to the paper.

**AP3** Ran ABC analysis on data and appended results to manuscript

**References**

Calvo, A., Baumgardner, D., Castro, A., Fernández-González, D., Vega-Maray, A., Valencia-Barrera, R., Oduber, F., Blanco-Alegre, C., and Fraile, R. (2018). Daily behavior of urban fluorescing aerosol particles in northwest spain. *Atmospheric Environment*, 184:262–277.

Crawford, I., Ruske, S., Topping, D., and Gallagher, M. (2015). Evaluation of hierarchical agglomerative cluster analysis methods for discrimination of primary biological aerosol. *Atmospheric Measurement Techniques*, 8(11):4979–4991.

Crotzer, V. and Levetin, E. (1996). The aerobiological significance of smut spores in tulsa, oklahoma. *Aerobiologia*, 12(1):177–184.

Gillbro, T. and Cogdell, R. J. (1989). Carotenoid fluorescence. *Chemical Physics Letters*, 158(3-4):312–316.

Harrison, R. M., Jones, A. M., Biggins, P. D., Pomeroy, N., Cox, C. S., Kidd, S. P., Hobman, J. L., Brown, N. L., and Beswick, A. (2005). Climate factors influencing bacterial count in background air samples. *International journal of biometeorology*, 49(3):167–178.

Healy, D. A., O'Connor, D. J., Burke, A. M., and Sodeau, J. R. (2012). A laboratory assessment of the waveband integrated bioaerosol sensor (wibs-4) using individual samples of pollen and fungal spore material. *Atmospheric environment*, 60:534–543.

Hernandez, M., Perring, A. E., McCabe, K., Kok, G., Granger, G., and Baumgardner, D. (2016). Chamber catalogues of optical and fluorescent signatures distinguish bioaerosol classes. *Atmospheric Measurement Techniques*, 9(7).

Kang, D.-Y., Son, M.-S., Eum, C.-H., Kim, W.-S., and Lee, S.-H. (2007). Size determination of pollens using gravitational and sedimentation field-flow fractionation. *Bulletin of the Korean Chemical Society*, 28(4):613–618.

Kaye, P. H., Stanley, W., Hirst, E., Foot, E., Baxter, K., and Barrington, S. (2005). Single particle multi-channel bio-aerosol fluorescence sensor. *Optics express*, 13(10):3583–3593.

Khaneja, R., Perez-Fons, L., Fakhry, S., Baccigalupi, L., Steiger, S., To, E., Sandmann, G., Dong, T., Ricca, E., Fraser, P., et al. (2010). Carotenoids found in bacillus. *Journal of applied microbiology*, 108(6):1889–1902.

Pöhlker, C., Huffman, J., and Pöschl, U. (2012). Autofluorescence of atmospheric bioaerosols–fluorescent biomolecules and potential interferences. *Atmospheric Measurement Techniques*, 5(1):37–71.

Pöhlker, C., Huffman, J. A., Förster, J.-D., and Pöschl, U. (2013). Autofluorescence of atmospheric bioaerosols: spectral fingerprints and taxonomic trends of pollen. *Atmospheric Measurement Techniques*, 6(12):3369–3392.

Savage, N. J., Krentz, C. E., Könemann, T., Han, T. T., Mainelis, G., Pöhlker, C., and Huffman, J. A. (2017). Systematic characterization and fluorescence threshold strategies for the wideband integrated bioaerosol sensor (wibs) using size-resolved biological and interfering particles. *Atmospheric Measurement Techniques*, 10(11):4279–4302.

Tong, Y. and Lighthart, B. (1997). Solar radiation is shown to select for pigmented bacteria in the ambient outdoor atmosphere. *Photochemistry and photobiology*, 65(1):103–106.